# Does Example Selection for In-Context Learning Amplify the Biases of Large Language Models?

## Abstract

In-context learning (ICL) has proven to be adept at adapting large language models (LLMs) to downstream tasks without parameter updates, based on a few demonstration examples. Prior work has found that the ICL performance is susceptible to the selection of examples in prompt and made efforts to stabilize it. However, existing example selection studies ignore the ethical risks behind the examples selected, such as gender and race bias. In this work, we first construct a new sentiment classification dataset —*EEC-paraphrase*, designed to better capture and evaluate the biases of LLMs. Then, through further analysis, we discover that ❶ **example selection with high accuracy does not mean low bias; ❷ example selection for ICL amplifies the biases of LLMs; ❸ example selection contributes to spurious correlations of LLMs.** Based on the above observations, we propose the ***Re**mind with **B**ias-aware **E**mbedding* (**ReBE**), which removes the spurious correlations through contrastive learning and obtains bias-aware embedding for LLMs based on prompt tuning. Finally, we demonstrate that ReBE effectively mitigates biases of LLMs without significantly compromising accuracy and is highly compatible with existing example selection methods. *The implementation code is available at https://anonymous.4open.science/r/ReBE-1D04.*

## 1 Introduction

Although large language models (LLMs) have demonstrated impressive capabilities, efficiently deploying them into downstream tasks remains challenging (Mosbach et al., 2023; Liu et al., 2022a). Among existing solutions, in-context learning (ICL) has proven adept at adapting LLMs to downstream tasks without parameter updates, using only a few demonstration examples (Brown et al., 2020). Compared to fine-tuning (Ziegler et al., 2019), ICL is more flexible and suitable for few-shot scenarios. **In the setting of ICL, examples included in the prompt are the only source for LLMs to learn the task context information (e.g., the answer format), thus attracting considerable attention.** As the research deepened, researchers found that examples selected randomly from the training set led to high variance in performance (Liu et al., 2022b), so numerous example selection methods have been proposed to stabilize the performance of ICL (Gonen et al., 2023; Gupta et al., 2023).

Since LLMs may spread biases learned from the training set during decision-making or user interaction, potentially causing severe harm to society, the biases of LLMs have always attracted significant attention (Liu et al., 2024b; Gupta et al., 2024; Guo et al., 2022). Although not entirely equivalent to social biases, it has been shown that LLMs exhibit stronger cognitive biases (Lin & Ng, 2023), such as position bias (Zhao et al., 2021) and token bias (Zheng et al., 2024), when fed with specific prompts. Similarly, because the example selection method determines the content of the ICL prompt, it is natural to ask: **Does example selection for ICL amplify the biases of LLMs?** It is undoubtedly unacceptable for LLMs to preserve or even exacerbate biases when using ICL to deploy LLMs to downstream tasks. However, **existing example selection studies ignore the ethical risks behind the examples selected**, such as gender and race bias.

To explore the impact of example selection on bias, we conduct an empirical analysis by evaluating the accuracy and biases of LLMs on a sentiment classification dataset —*EEC-paraphrase*, which

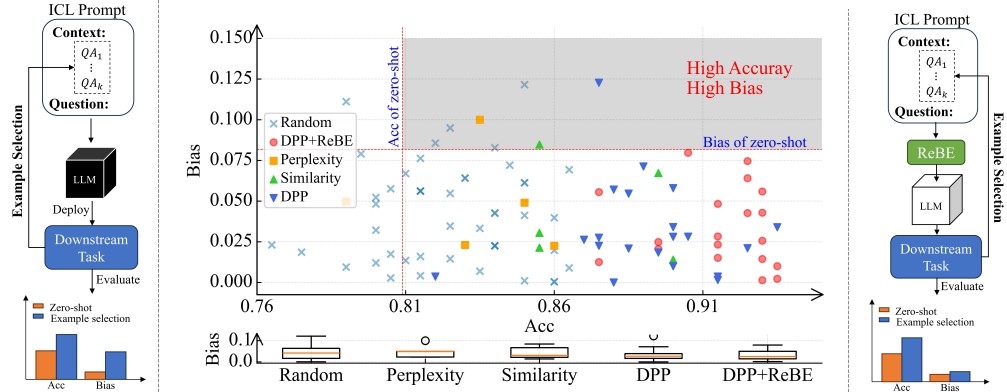

Figure 1: The **central** scatter figure plots gender bias and accuracy of `OPT-13B` under various example selection baselines. The horizontal and vertical red dashed lines represent mean accuracy and maximum bias (AvgGF) of `OPT-13B` under zero-shot, respectively. The **left** subfigure shows the pipeline for adapting LLMs to downstream tasks using ICL. The **right** subfigure illustrates the pipeline using our debiasing method, ReBE. The box plot at the **bottom** depicts the gender bias distribution of `OPT-13B` under various baselines.

we build on *Equity Evaluation Corpus (EEC)* (Kiritchenko & Mohammad, 2018) but with more complex and natural sentences (More details in Section 3). Considering the generality of the findings, our experiments include eight LLMs and four example selection baselines: Random-based, Similarity-based (Liu et al., 2022b), Perplexity-based (Gonen et al., 2023) and Determinantal Point Processes (DPP)-based (Ye et al., 2023). We use random seeds to sample the *EEC-paraphrase* to construct the few-shot training sets and have collected the bias and accuracy results of baselines under various random seeds. Therefore, we emphasize that the data points of example selection baselines in Figure 1 are evaluation results under different random seeds. According to Figure 1, each example selection baseline has points in the grey area marked as "high accuracy and high bias", indicating that **example selection with high accuracy does not mean low bias**.

To observe the impact of example selection on biases compared to the case without ICL, we have also collected the experiment results of zero-shot under various random seeds and plotted the red dashed line "Bias of zero-shot" with the maximum bias value in Figure 1. The data points above the horizontal red dashed line in Figure 1 exhibit higher gender bias than zero-shot, indicating that **example selection for ICL does amplify the bias of LLMs**. According to the results in Section 3.3, we further find that example selection amplifies the **maximum bias value**, worsening unfair situations. The maximum bias value refers to the highest bias among results measured under various random seeds using the same example selection method. To uncover why example selection amplifies the biases, based on the MaxTG and MaxFG metrics (Table 1), we observe that LLMs using ICL exhibit **spurious correlations**. Spurious correlations refer to undesired or unstable correlations learned by LLMs from the training set, which may introduce unintended biases (Albuquerque et al., 2024). Typical spurious correlations of LLMs include stereotypes such as "He is a doctor; she is a nurse." Furthermore, it is generally believed that the LLM's biases come from its parameter knowledge and the input prompt. By excluding the impact of LLM parameters, we find that **example selection contributes to spurious correlations of LLMs**.

The above observations highlight that example selection for ICL truly amplifies the biases of LLMs. In order to mitigate the social biases of adapting LLMs to downstream tasks through ICL, we propose the *Remind with Bias-aware Embedding* (ReBE), which curbs biases of LLMs by prefixing the bias-aware embedding into the prompt. Besides, we design the bias-contrastive loss based on contrastive learning to remove spurious correlations and obtain the bias-aware embedding through prompt tuning (More details in Section 4). To demonstrate the effectiveness of ReBE, we conduct extensive experiments and the results in Section 5 show that ReBE reduces the maximum bias value without compromising the accuracy and is well compatible with existing example selection methods. In sum, we try to fill the gap in exploring the ethical risks of example selection, which is essential

for deploying LLMs into downstream tasks using ICL. The overall contributions are summarized as follows:

- To the best of our knowledge, **we are the first** to discover the bias risks of example selection for ICL, especially the findings: ❶ Example selection with high accuracy does not mean low bias; ❷ Example selection for ICL amplifies the biases of LLMs; ❸ Example selection contributes to spurious correlations of LLMs.

- We construct a new sentiment classification dataset —*EEC-paraphrase*, which can better identify and evaluate gender and race bias of LLMs in ICL. More specifically, sentences in *EEC-paraphrase* are more complex and natural than in *EEC*.

- To alleviate the bias amplification of example selection, we propose the **Re**mind with **B**ias-aware **E**mbedding (**ReBE**), which removes spurious correlations by minimizing the bias-contrastive loss while preserving the advantages of ICL through prompt tuning.

- We conduct extensive experiments to validate the effectiveness of ReBE, including four LLMs and four example selection baselines.

## 2 Preliminaries

### 2.1 Example Selection for ICL

Given a test input $x_{test}$, ICL enables the language model $\mathcal{M}$ to learn how to generate $y_{test}$ from just a few examples in the context $C$. The above process can be formulated as:

$$\hat{y} = \arg\max_{y \in \mathcal{Y}} p_{\mathcal{M}}(y|C, x_{test}), \tag{1}$$

where $\hat{y}$ is the prediction, $\mathcal{Y}$ is the label set, and $p_{\mathcal{M}}(y|C, x_{test})$ represents the probability that $\mathcal{M}$ generates $y$ with context $C$ and $x_{test}$ as input. For a task with training set $\mathcal{D} = \{(x_i, y_i)\}_{i=1}^N$, if context $C$ contains $k$ examples ($k$-shot prompt), then $C = \{(x_1, y_1), (x_2, y_2), ..., (x_k, y_k)\} \subset \mathcal{D}$.

Among current studies (Iter et al., 2023; Yang et al., 2023), example/demonstration selection and example/demonstration retriever are interchangeable. To avoid confusion, we use the term *example selection* throughout this paper. Since the performance of $\mathcal{M}$ depends on context $C$, we need to select examples $(x_i, y_i)$ to minimize the total loss on the test set $(\mathbf{x}_{test}, \mathbf{y}_{test})$, which could be formulated as the following problem:

$$C^* = \arg\min_{C \subset \mathcal{D}} \mathcal{L}_{\mathcal{M}}(\hat{\mathbf{y}}, \mathbf{y}_{test}), \tag{2}$$

where $\hat{\mathbf{y}} = \{\arg\max_{y \in \mathcal{Y}} p_{\mathcal{M}}(y|C, x_{test})\}, x_{test} \in \mathbf{x}_{test}$, and $C^*$ is the desired sample subset of example selection methods.

### 2.2 Contrastive Learning

Contrastive learning aims to obtain representation by maximizing the similarity between related samples and minimizing the similarity between unrelated samples, simultaneously. Although originating from self-supervised learning, contrastive learning also proves useful in supervised learning (Khosla et al., 2020; Chen et al., 2022). Given a training set $\mathcal{D} = \{(x_i, y_i)\}_{i=1}^N$ and its indexes set $\mathcal{I} = \{1, 2, ..., N\}$, define the $i$-th sample $x_i$ as an *anchor*, the contrastive loss for supervised tasks (Khosla et al., 2020) can be defined as:

$$\mathcal{L}_{sup} = -\sum_{i \in \mathcal{I}} \frac{1}{|\mathcal{P}(i)|} \sum_{p \in \mathcal{P}(i)} log \frac{exp(z_i \cdot z_p/\tau)}{\sum_{a \in \mathcal{A}(i)} exp(z_i \cdot z_a/\tau)}, \tag{3}$$

where $z_i$ is the normalized representation of anchor $x_i$, $\mathcal{P}(i) = \{p \in \mathcal{A}_i : y_p = y_i\}$ is the index set of *positive* samples. $\mathcal{A}_i = \mathcal{I} \setminus \{i\}$ is the index set of contrastive samples that removes $i$ from set $\mathcal{I}$ and $\tau$ is the temperature parameter. Constructing sensible $\mathcal{P}(i)$ and $\mathcal{A}(i)$ is vital to utilizing the contrastive learning framework.

# 3 EXPLORE THE IMPACT OF EXAMPLE SELECTION ON LLM BIASES

## 3.1 DATASET AND MODELS

**Dataset** To better capture and evaluate the gender and race bias of LLMs, we construct a new sentiment classification dataset —*EEC-paraphrase*. Given a sentence in the template `<Person> feels <emotional word>.`, LLMs are asked to identify the sentiment contained in the sentence. By replacing `<Person>` with first names (e.g., Alonzo and Alan) or pronouns (e.g., she and he) associated with specific demographic group, *EEC-paraphrase* includes 8,640 English sentences with gender and race attributes.

*EEC-paraphras*e is built through paraphrasing sentences in the *Equity Evaluation Corpus (EEC)* (Kiritchenko & Mohammad, 2018) by `GPT-3.5-Turbo`. Compared with *EEC*, sentences in *EEC-paraphrase* are more complex and natural, closer to the actual scenario (The quality validation is available in Appendix A.). Besides, to simulate the few-shot scenario, we build a *train400-dev200* dataset by randomly sampling 400 sentences for the training set and 200 sentences for the development set from the *EEC-paraphrase*.

**Language Models** To guarantee the reliability of our findings, we conduct experiments on eight LLMs, including `LlaMA-2-7/13/70B`, `OPT-6.7/13/30B`, `GPT-J-6B` and `GPT-neo-2.7B`. LLMs with various parameter sizes but within the same series facilitate our analysis of the effects of parameter quantities.

Table 1: Bias metrics for sentiment classification.

| Metric | Formula |
|---|---|
| Average Group Fairness | $\mathbf{AvgGF} = \big|P(\hat{Y}=Y|S=s_1) - P(\hat{Y}=Y|S=s_2)\big|$ |
| Maximum TPR Gap | $\mathbf{MaxTG} = \max_{y \in \mathcal{Y}} \big|P(\hat{Y}=y|Y=y \cap S=s_1) - P(\hat{Y}=y|Y=y \cap S=s_2)\big|$ |
| Maximum FPR Gap | $\mathbf{MaxFG} = \max_{y,\hat{y} \in \mathcal{Y}, \hat{y} \neq y} \big|P(\hat{Y}=\hat{y}|Y=y \cap S=s_1) - P(\hat{Y}=\hat{y}|Y=y \cap S=s_2)\big|$ |

\* $s_1$ and $s_2$ correspond to different demographic groups.

To further validate the generalizability of our findings, we evaluate LLMs on the toxicity detection task using the [1]Jigsaw dataset. The results are available in Appendix F.

## 3.2 BIAS METRICS AND BASELINES

Since the output of LLMs is not numerical value but sentences containing the judgment result, we evaluate the prediction's accuracy by comparing the semantic similarity between the answer and options.

**Metrics** Drawing on fairness metrics of machine learning (Mehrabi et al., 2021) and natural language processing (Czarnowska et al., 2021), we summarize three representative bias metrics in Table 1, which adapts to the sentiment classification task. The basis for selecting metric is whether it can reflect the unfairness or stereotypes of different groups in various sentiments. See Appendix B for a detailed explanation of metrics.

**Baselines** We select four example selection methods as baselines to study the impact of example selection on the biases of LLMs. *Random-based* example selection refers to randomly choosing examples from the training set to form a few-shot prompt. *Similarity-based* (Liu et al., 2022b) and *perplexity-based* example selection (Gonen et al., 2023) picks the top-k examples based on semantic similarity and perplexity of example, respectively. *Determinantal Point Processes (DPP)-based* example selection (Ye et al., 2023) uses DPP to consider two properties simultaneously when selecting examples.

---

[1]Jigsaw unintended bias in toxicity classification

### 3.3 IMPACTS OF EXAMPLE SELECTION ON BIAS OF LLMS

Although example selection aims to stabilize the performance of LLMs using ICL, inappropriate examples selected may also mislead LLMs. We assess the change in LLM biases when using example selections for ICL compared to zero-shot. **Figure 2 illustrates the differences in the maximum and mean bias values between random-based example selection and zero-shot.** The comparisons of the remaining example selection baselines are available in the Appendix C.1. It is evident that, although example selections reduce the mean bias value, the LLMs tested exhibit varying degrees of increase in the maximum gender or race bias value with random-based example selection for ICL. In other words, **example selection for ICL amplifies the biases of LLM**, increases the fluctuation of biases and exacerbates the unfair risks. Besides, the maximum bias values among LLMs for each baseline are highlighted in Table 2 and are significantly higher than the mean values.

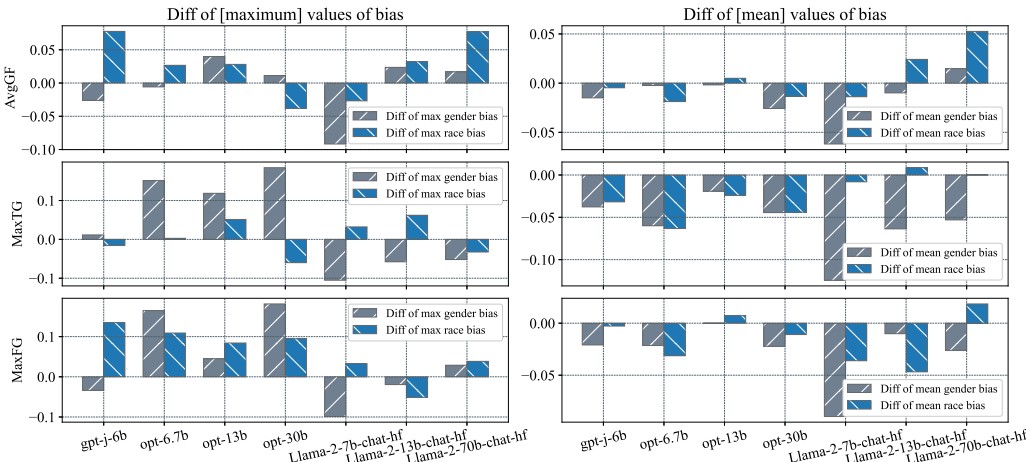

Figure 2: The impacts of random-based example selection on biases of LLMs. The bar value is calculated by Diff=Bias$_{random}$-Bias$_{zero-shot}$.

Table 2: Accuracy and gender bias of LLMs under four example selection baselines.

| | | GPT-J-6B | GPT-neo-2.7B | OPT-6.7B | OPT-13B | OPT-30B | Llama-2-7B | Llama-2-13B | Llama-2-70B |
|---|---|---|---|---|---|---|---|---|---|
| **Random** | Acc$_{(Min)}$ | 0.84$_{(0.80)}$ | 0.77$_{(0.58)}$ | 0.81$_{(0.67)}$ | 0.82$_{(0.72)}$ | 0.84$_{(0.76)}$ | 0.86$_{(0.81)}$ | 0.87$_{(0.83)}$ | 0.86$_{(0.82)}$ |
| | AvgGF$_{(Max)}$ | 0.04$_{(0.08)}$ | 0.04$_{(\mathbf{0.13})}$ | 0.04$_{(\mathbf{0.13})}$ | 0.04$_{(0.12)}$ | 0.04$_{(\mathbf{0.13})}$ | 0.03$_{(0.08)}$ | 0.04$_{(0.09)}$ | 0.04$_{(0.09)}$ |
| | MaxTG$_{(Max)}$ | 0.15$_{(0.29)}$ | 0.14$_{(0.31)}$ | 0.18$_{(\mathbf{0.47})}$ | 0.17$_{(0.38)}$ | 0.17$_{(\mathbf{0.47})}$ | 0.11$_{(0.22)}$ | 0.14$_{(0.25)}$ | 0.17$_{(0.30)}$ |
| | MaxFG$_{(Max)}$ | 0.17$_{(0.26)}$ | 0.20$_{(0.39)}$ | 0.20$_{(\mathbf{0.46})}$ | 0.19$_{(0.34)}$ | 0.19$_{(\mathbf{0.46})}$ | 0.13$_{(0.22)}$ | 0.14$_{(0.21)}$ | 0.17$_{(0.30)}$ |
| **Perplexity** | Acc$_{(Min)}$ | 0.83$_{(0.72)}$ | 0.82$_{(0.82)}$ | 0.85$_{(0.81)}$ | 0.83$_{(0.79)}$ | 0.86$_{(0.85)}$ | 0.865$_{(0.8)}$ | 0.87$_{(0.84)}$ | 0.86$_{(0.85)}$ |
| | AvgGF$_{(Max)}$ | 0.09$_{(\mathbf{0.15})}$ | 0.08$_{(0.08)}$ | 0.04$_{(0.09)}$ | 0.05$_{(\mathbf{0.10})}$ | 0.05$_{(0.09)}$ | 0.03$_{(0.04)}$ | 0.04$_{(0.07)}$ | 0.05$_{(0.08)}$ |
| | MaxTG$_{(Max)}$ | 0.23$_{(\mathbf{0.38})}$ | 0.18$_{(0.18)}$ | 0.21$_{(\mathbf{0.35})}$ | 0.22$_{(0.32)}$ | 0.20$_{(0.35)}$ | 0.18$_{(0.33)}$ | 0.17$_{(0.28)}$ | 0.20$_{(0.27)}$ |
| | MaxFG$_{(Max)}$ | 0.24$_{(\mathbf{0.50})}$ | 0.24$_{(\mathbf{0.50})}$ | 0.17$_{(0.31)}$ | 0.27$_{(0.46)}$ | 0.17$_{(0.22)}$ | 0.14$_{(0.28)}$ | 0.14$_{(0.19)}$ | 0.17$_{(0.22)}$ |
| **Similarity** | Acc$_{(Min)}$ | 0.92$_{(0.88)}$ | 0.85$_{(0.82)}$ | 0.84$_{(0.82)}$ | 0.87$_{(0.86)}$ | 0.90$_{(0.86)}$ | 0.93$_{(0.90)}$ | 0.92$_{(0.90)}$ | 0.89$_{(0.87)}$ |
| | AvgGF$_{(Max)}$ | 0.03$_{(0.06)}$ | 0.03$_{(\mathbf{0.09})}$ | 0.03$_{(0.05)}$ | 0.04$_{(\mathbf{0.09})}$ | 0.02$_{(0.04)}$ | 0.03$_{(0.08)}$ | 0.03$_{(0.04)}$ | 0.04$_{(0.07)}$ |
| | MaxTG$_{(Max)}$ | 0.13$_{(0.28)}$ | 0.19$_{(\mathbf{0.30})}$ | 0.12$_{(0.22)}$ | 0.21$_{(\mathbf{0.38})}$ | 0.13$_{(0.30)}$ | 0.16$_{(0.27)}$ | 0.16$_{(0.25)}$ | 0.16$_{(0.23)}$ |
| | MaxFG$_{(Max)}$ | 0.14$_{(0.20)}$ | 0.16$_{(0.19)}$ | 0.15$_{(\mathbf{0.31})}$ | 0.17$_{(\mathbf{0.37})}$ | 0.11$_{(0.18)}$ | 0.13$_{(0.21)}$ | 0.17$_{(0.27)}$ | 0.13$_{(0.16)}$ |
| **DPP** | Acc$_{(Min)}$ | 0.93$_{(0.89)}$ | 0.89$_{(0.83)}$ | 0.87$_{(0.79)}$ | 0.89$_{(0.82)}$ | 0.91$_{(0.86)}$ | 0.94$_{(0.90)}$ | 0.93$_{(0.91)}$ | 0.90$_{(0.85)}$ |
| | AvgGF$_{(Max)}$ | 0.03$_{(0.06)}$ | 0.03$_{(0.07)}$ | 0.04$_{(\mathbf{0.11})}$ | 0.03$_{(\mathbf{0.12})}$ | 0.02$_{(0.06)}$ | 0.02$_{(0.06)}$ | 0.02$_{(0.05)}$ | 0.03$_{(0.08)}$ |
| | MaxTG$_{(Max)}$ | 0.12$_{(0.28)}$ | 0.13$_{(\mathbf{0.28})}$ | 0.14$_{(0.27)}$ | 0.13$_{(\mathbf{0.38})}$ | 0.11$_{(0.28)}$ | 0.10$_{(0.18)}$ | 0.10$_{(0.25)}$ | 0.12$_{(0.25)}$ |
| | MaxFG$_{(Max)}$ | 0.10$_{(0.17)}$ | 0.13$_{(0.24)}$ | 0.14$_{(\mathbf{0.27})}$ | 0.12$_{(\mathbf{0.38})}$ | 0.10$_{(0.18)}$ | 0.09$_{(0.22)}$ | 0.09$_{(0.17)}$ | 0.12$_{(0.21)}$ |

[1] Avg$_{(Min)}$ are the largest two values in AvgGF; Avg$_{(Min)}$ are the largest two values in MaxTG and MaxFG.

### 3.4 SPURIOUS CORRELATIONS OBSERVED WITH MaxTG AND MaxFG

As seen from Figure 3, we visualize the confusion matrices of `OPT-6.7B`, which has the biggest fluctuation of MaxTG (0.47) and MaxFG (0.46) in Table 2. With the help of Figure 3, we can further analyze the reasons that cause MaxTG and MaxFG to increase. For MaxTG, by comparing the first two sub-figures of Figure 3 by row, the proportion of sadness sentences correctly predicted in the female group (0.88) is higher than in the male group (0.42), which is consistent with the finding of Plaza-del Arco et al. (2024). Likewise, for MaxFG, by comparing the first two sub-figures of Figure 3 by column, more sentences with *sadness* labels are incorrectly predicted as *fear* in the male group (0.54) than in the female (0.08). We believe the disparity —where sentences labelled as *sadness* containing male pronouns are more easily misjudged as *fear* than those with female pronouns —occurs because the sentiment analysis criteria of LLMs may be influenced by words other than emotional ones, leading to **spurious correlations**. Although it has been proven that spurious correlations exist in LLMs, we are unsure whether the example selection methods for ICL contribute to these spurious correlations.

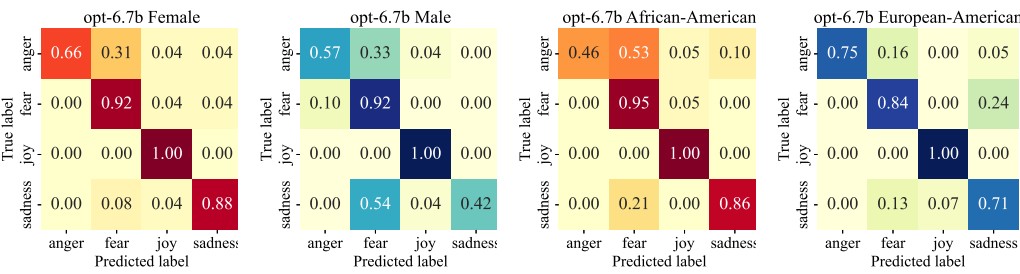

Figure 3: Confusion matrix heatmaps of `OPT-6.7B`.

Two factors affect the biases of LLMs: the LLM parameters and the input prompt. The former refers to biased knowledge that LLMs acquire during pre-training, which we call *native bias*. To isolate the influence of native bias, we use **null (content-free) prompts** (Lin & Ng, 2023; Zhao et al., 2021) to observe the tendency of LLMs parameters. More specifically, the null prompt fills the `<Position>` in the template with demographic-related words, leaves the emotional word empty, and tests the probability of LLMs' prediction for each sentiment label. Combined with the spurious correlation between *male* and *fear* in Figure 3, the fear-label tendency of `OPT-6.7B` in Figure 4 is nearly identical for *female* and *male*, indicating that spurious correlations are not caused entirely by the LLM parameters and **example selection contributes to spurious correlations**.

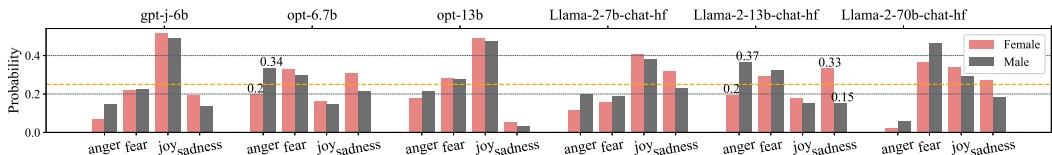

Figure 4: The native bias of LLMs over various sentiment labels.

## 4 ReBE: REMIND WITH BIAS-AWARE EMBEDDING

To retain the accuracy and flexibility of ICL while reducing bias, we propose the ReBE, which removes spurious correlations based on contrastive learning and reminds LLMs of fairness with bias-aware embedding.

### 4.1 THE OVERVIEW OF ReBE

As shown in Figure 5, using $(x, y, s)$ as input, ReBE obtains bias-aware embedding by minimizing the bias-contrastive loss during training. Here, $x$, $y$, and $s$ correspond to the task's sample, label,

and demographic attribute. With the help of prompt tuning, ReBE avoids updating the original parameters of LLM $\mathcal{M}$, retaining the flexibility of ICL. Besides, to effectively remove spurious correlations, contrastive learning is introduced to construct the bias-contrastive loss. The verbalizer (Cui et al., 2022) converts representations $\{z_1, z_2, ..., z_k\}$ to predicted labels $\{joy, anger, ...\}$ used in the downstream task.

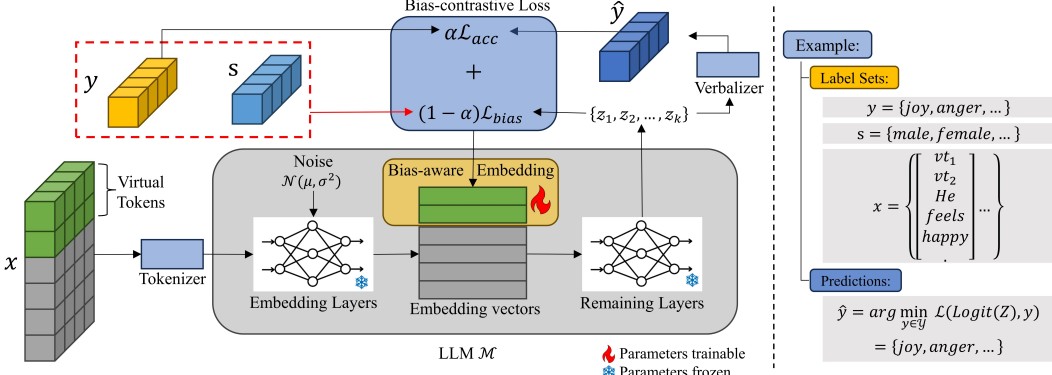

Figure 5: The overview of ReBE. The left side of the figure depicts the framework of ReBE, including the input $(x, y, s)$ and the process of obtaining the Bias-aware Embedding. The right side of the figure is an example of inputs and output of ReBE.

## 4.2 BIAS-AWARE EMBEDDING

**Prompt Tuning** (Lester et al., 2021; Gu et al., 2022) is a soft (continuous) prompt construction and parameter-efficient tuning method for LLMs, which generally searches for the best ICL prompt in the semantic space via back-propagation. By adding virtual (pseudo) tokens to the prompt of LLMs, prompt tuning obtains trainable parameters after the embedding processing. We name trainable parameters in the prompt tuning for LLMs debiasing as **Bias-aware Embedding**. It should be noted that virtual tokens have no real meaning and only serve as placeholders. Besides, the contexts of prompt during prompt tuning are constructed based on the example selection method.

To better explain the generation of bias-aware embedding, we take the sentiment classification task in Figure 5 as an example. Represent the sentence as $x = [v1][v2][He][feels][happy][.]$, where $[v_i]$ is the virtual token. After tokenization and embedding processing, bias-aware embedding becomes part of embedding vectors, which are fed into the remaining neural network layers. Representing the number of virtual tokens $[v_i]$ as $n_{vr}$, and the dimension of LLM feature vectors as $n_{feats}$, the number of trainable parameters (bias-aware embedding) can be calculated as $n_{vr} \times n_{feats}$. All original parameters of LLM are frozen and are not involved in the training process described above. Since prompt tuning has been found to be unstable during training (Chen et al., 2023a), we add Gaussian noise to help the training, which is a common solution (Wu et al., 2022; Pecher et al., 2024).

Through back-propagation and gradient descent, the trainable parameters are updated to minimize the loss and obtain bias-aware embedding, which is then saved in the embedding table of LLM. According to the corresponding virtual tokens, bias-aware embedding is integrated into the embedding vectors during inference.

## 4.3 BIAS-CONTRASTIVE LOSS

Acquiring bias-aware embedding requires a well-designed loss function to guide the training. Given a training set $\mathcal{D} = \{(x_i, y_i)\}_{i=1}^{N}$ and its indexes set $\mathcal{I} = \{1, 2, ..., N\}$, define $z_i$ as the normalized representation of sample $x_i$. To better mitigate biases in the representation of LLM, we first design the bias-contrastive loss $\mathcal{L}_{bias}$ based on SupCon (Khosla et al., 2020) loss as follows:

$$\mathcal{L}_{bias} = \frac{1}{N} \sum_{i \in \mathcal{I}} \frac{1}{|\mathcal{P}(i)|} \sum_{j \in \mathcal{P}(i)} -log \frac{exp(z_i \cdot z_j / \tau)}{\sum_{k \in \mathcal{A}(i)} exp(z_i \cdot z_k / \tau)}, \quad (4)$$

where $\mathcal{P}(i) = \{j \in \mathcal{I} : y_j = y_i, s_j \neq s_i\}$, represents the set of indexes of examples with the same label and different demographic attribute $s_j$ as $z_i$. Conversely, $\mathcal{A}(i) = \{k \in \mathcal{I} : y_k \neq y_i, s_k = s_i\}$, represents the set of indexes of examples with the different label and same demographic attribute as $z_i$. $\tau$ is the temperature parameter of contrastive learning.

On the other hand, to retain the accuracy of ICL, we introduce the loss $\mathcal{L}_{acc}$ based on cross-entropy loss. Following the convention, we define the $\mathcal{L}_{acc}$ as:

$$\mathcal{L}_{acc} = \frac{1}{N} \sum_{i \in \mathcal{I}} -\log \frac{exp(p_i)}{\sum_{y \in \mathcal{Y}} exp(p_i^y)}, \quad (5)$$

where $p_i$ is the probability that $z_i$ is predicted to be the ground-truth label, $p_i^y$ is the probability that $z_i$ is predicted to be the label $y$, and label set $\mathcal{Y} = \{joy, anger, sadness, fear\}$.

Finally, we obtain bias-aware embedding by minimizing the weighted sum of the above two objectives: $\mathcal{L}_{total} = \alpha\mathcal{L}_{acc} + (1 - \alpha)\mathcal{L}_{bias}$, where $\alpha$ is the parameter that balances the accuracy and fairness. As shown in Figure 5, the total loss $\mathcal{L}_{total}$ is used to optimize the bias-aware embedding via back-propagation.

Table 3: Gender bias and accuracy of LLMs under example selections after debiasing.

| | | Acc↑ | AvgGF↓ | MaxTG↓ | MaxFG↓ | Acc↑ | AvgGF↓ | MaxTG↓ | MaxFG↓ |
|---|---|---|---|---|---|---|---|---|---|
| **Random** | Max | **GPT-neo-2.7B** 0.083(-0.044) | 0.260(-0.055) | 0.319(-0.067) | **OPT-6.7B** 0.086(-0.042) | 0.322(-0.146) | 0.447(-0.018) |
| | Avg | 0.828(+0.150) | 0.035(-0.000) | 0.135(-0.008) | 0.156(-0.042) | 0.781(-0.027) | 0.034(-0.011) | 0.151(-0.029) | 0.191(-0.006) |
| **Perplexity** | Max | **GPT-J-6B** 0.064(-0.024) | 0.350(-0.035) | 0.381(-0.122) | **OPT-13B** 0.113(+0.013) | 0.300(-0.021) | 0.301(-0.157) |
| | Avg | 0.829(-0.002) | 0.064(-0.024) | 0.171(-0.060) | 0.164(-0.079) | 0.828(-0.005) | 0.058(+0.009) | 0.201(-0.019) | 0.172(-0.096) |
| **Similarity** | Max | **GPT-neo-2.7B** 0.053(-0.036) | 0.267(-0.033) | 0.167(-0.026) | **OPT-13B** 0.062(-0.022) | 0.333(-0.050) | 0.283(-0.083) |
| | Avg | 0.871(+0.024) | 0.031(-0.003) | 0.140(-0.047) | 0.132(-0.032) | 0.896(+0.024) | 0.032(-0.012) | 0.181(-0.028) | 0.167(-0.008) |
| **DPP** | Max | **OPT-6.7B** 0.073(-0.037) | 0.250(-0.023) | 0.247(-0.026) | **OPT-13B** 0.080(-0.043) | 0.267(-0.117) | 0.167(-0.217) |
| | Avg | 0.874(+0.009) | 0.033(-0.003) | 0.120(-0.022) | 0.122(-0.021) | 0.918(+0.027) | 0.033(+0.001) | 0.120(-0.008) | 0.100(-0.021) |

[1] Red subscript indicates that the metric increases after debiasing, and blue subscript indicates that the metric decreases after debiasing.

## 5 EXPERIMENTAL RESULTS

### 5.1 RESULTS AFTER DEBIASING BY REBE

To validate the few-shot performance of ReBE, we conduct debiasing experiments on a training set of 400 samples and a test set of 200 samples, split from the *EEC-paraphrase*. According to results in Table 2, we select the two LLMs with the largest AvgGF in each baseline to eliminate the gender bias. The experimental results of race bias are available in Appendix D.2. Due to hardware limitations, we exclude OPT-30b and Llama-2-70b from the choices. We implement the ReBE based on the Huggingface PEFT library and previous work (Nguyen & Wong, 2023).

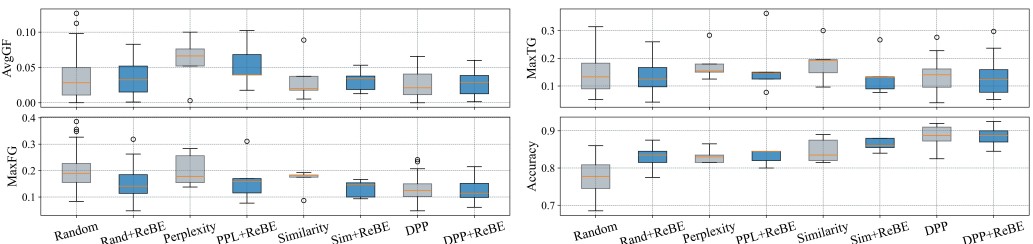

Figure 6: The accuracy and gender bias comparison of GPT-neo-2.7B under four example selection baselines before and after debiasing.

As shown by the blue subscripts in Table 3, the average gender bias of most LLMs decreases after debiasing by ReBE, which works for all example selection baselines. Concerning the issue that example selection may amplify the maximum bias value, the "Max" row in Table 3 shows a significant reduction in maximum bias. In addition, Figure 6 more intuitively shows the changes in accuracy, AvgGF, MaxTG and MaxFG of GPT-neo-2.7B before and after debiasing. The variances of the

three biases all decrease, resulting in a more concentrated distribution, indicating improved stability of the bias. In addtion, according to Table 3, the sentiment classification accuracy of LLMs is not significantly affected after using ReBE. The above experimental results demonstrate that **ReBE meets the requirement of reducing bias without significantly compromising the accuracy**. More importantly, the results in Table 3 and Figure 6 demonstrate that **ReBE is compatible with existing example selection methods**. By combining example selection with ReBE, it is possible to achieve high accuracy and low bias of LLMs.

## 5.2 ABLATION STUDY

To further demonstrate that the reduction in bias results from the $\mathcal{L}_{bias}$ rather than improved accuracy, we conduct ablation studies using the $\mathcal{L}_{acc}$ and $\mathcal{L}_{bias}$ to replace the $\mathcal{L}_{total}$ to train the GPT-J-6B, respectively. As shown in Table 4, the maximum values of AvgGF and MaxTG of $\mathcal{L}_{acc}$ are much higher than those of ReBE, even though the accuracy is slightly improved. In contrast, $\mathcal{L}_{bias}$ achieves lower bias but sacrifices accuracy. Therefore, $\mathcal{L}_{bias}$ is actually responsible for bias reduction, and $\mathcal{L}_{acc}$ guarantees accuracy.

Table 4: Experimental results of ablation study and parameter analysis of GPT-J-6B.

| | Accuracy↑ | | AvgGF↓ | | MaxTG↓ | | MaxFG↓ | |
|---|---|---|---|---|---|---|---|---|
| | Mean | Min | Mean | Max | Mean | Max | Mean | Max |
| **Original** | $0.84_{(\pm 1.7\%)}$ | **0.80** | $0.04_{(\pm 2.2\%)}$ | 0.084 | $\underline{0.15}_{(\pm 5.3\%)}$ | 0.295 | $\underline{0.17}_{(\pm 4.9\%)}$ | $\underline{0.264}$ |
| $\mathcal{L}_{acc}$ | $\mathbf{0.86}_{(\pm 2.3\%)}$ | 0.77 | $0.03_{(\pm 2.3\%)}$ | 0.089 | $\underline{0.13}_{(\pm 5.2\%)}$ | 0.292 | $\underline{0.14}_{(\pm 4.2\%)}$ | **0.250** |
| $\mathcal{L}_{bias}$ | $0.26_{(\pm 1.7\%)}$ | 0.25 | $\mathbf{0.02}_{(\pm 0.9\%)}$ | **0.049** | $\mathbf{0.02}_{(\pm 4.9\%)}$ | **0.196** | $\mathbf{0.03}_{(\pm 7.7\%)}$ | 0.327 |
| **ReBE** | $\underline{0.84}_{(\pm 2.2\%)}$ | $\underline{0.78}$ | $\underline{0.03}_{(\pm 2.2\%)}$ | $\underline{0.082}$ | $0.14_{(\pm 3.9\%)}$ | $\underline{0.221}$ | $0.18_{(\pm 4.5\%)}$ | 0.284 |
| $n$-virtual | $0.85_{(\pm 1.3\%)}$ | $0.80_{(\pm 1.9\%)}$ | $0.03_{(\pm 0.9\%)}$ | $0.09_{(\pm 2.1\%)}$ | $0.13_{(\pm 2.6\%)}$ | $0.26_{(\pm 5.7\%)}$ | $0.15_{(\pm 2.2\%)}$ | $0.25_{(\pm 5.6\%)}$ |
| **Order** | $0.83_{(\pm 0.5\%)}$ | $0.74_{(\pm 4.9\%)}$ | $0.03_{(\pm 1.9\%)}$ | $0.09_{(\pm 2.1\%)}$ | $0.13_{(\pm 0.7\%)}$ | $0.27_{(\pm 3.5\%)}$ | $0.17_{(\pm 1.3\%)}$ | $0.31_{(\pm 4.8\%)}$ |

## 5.3 BASELINE COMPARISON

Regarding baseline selection, although Hu et al. (2024) proposed Fairness via Clustering Genetic (FCG) algorithm, it cannot be applied to sentiment analysis or toxicity detection because it requires explicit feature vectors for clustering. Since there are no other debiasing methods specifically for ICL, we compare ReBE with two context augmentation methods: counterfactual context and gender-balanced context. See the Appendix G for details of these two methods. As shown in Table 5, compared with the counterfactual context and gender-balanced context method, ReBE is compatible with existing example selection methods and can achieve lower bias and higher accuracy.

Table 5: Gender bias of OPT-6.7B under various example selection methods

| | AvgGF↓ | | MaxTG↓ | | MaxFG↓ | | Accuracy↑ |
|---|---|---|---|---|---|---|---|
| | Mean | Max | Mean | Max | Mean | Max | |
| Random | $0.044_{(\pm 0.03)}$ | 0.129 | $0.180_{(\pm 0.09)}$ | 0.468 | $0.199_{(\pm 0.09)}$ | 0.465 | 0.81 |
| DPP | $0.036_{(\pm 0.03)}$ | 0.110 | $0.142_{(\pm 0.08)}$ | 0.273 | $0.144_{(\pm 0.06)}$ | 0.273 | 0.87 |
| Gender-balanced | $0.040_{(\pm 0.03)}$ | 0.132 | $0.174_{(\pm 0.08)}$ | 0.333 | $0.210_{(\pm 0.09)}$ | 0.417 | 0.80 |
| Counterfactual | $0.035_{(\pm 0.03)}$ | 0.125 | $0.145_{(\pm 0.07)}$ | 0.369 | $0.149_{(\pm 0.07)}$ | 0.369 | 0.77 |
| Random+ReBE | $0.034_{(\pm 0.02)}$ | 0.086 | $0.151_{(\pm 0.07)}$ | 0.322 | $0.191_{(\pm 0.08)}$ | 0.447 | 0.78 |
| DPP+ReBE | $\mathbf{0.033}_{(\pm 0.02)}$ | **0.073** | $\mathbf{0.120}_{(\pm 0.05)}$ | **0.250** | $\mathbf{0.122}_{(\pm 0.05)}$ | **0.247** | **0.87** |

## 5.4 PARAMETER ANALYSIS

To illustrate the influence of parameters on ReBE, we conduct the following parameter analysis. Detailed results are available in Appendix D.3.

$k$**-shot** refers to the number of examples in prompt of ICL. Since the coverage of examples affects the accuracy of ICL (Gupta et al., 2023), the value of $k$ should be large enough. However, redundant information caused by excessive examples may decline the performance of ICL. As shown in Figure 7, the accuracy of LLMs after debiasing increases with the rise in $k$, while the biases tend to decrease initially and then increase. Therefore, considering accuracy and biases, we choose $k = 18$ as our

experiment setting. The analysis of the impact of an increasing number of ICL examples is available in Appendix E.

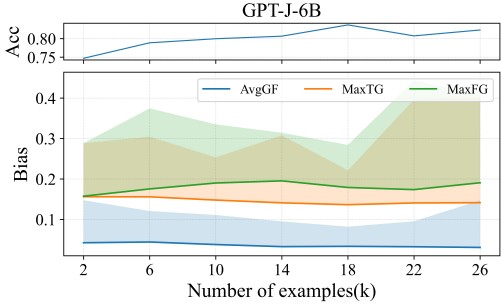 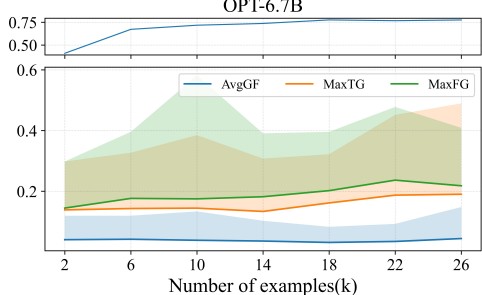

Figure 7: The accuracy and gender bias of LLM using ReBE under different $k$-shot.

$n$-**virtual** is the parameter of the prompt tuning, which refers to the number of virtual prompt tokens and decides the size of trainable parameters. ReBE needs enough parameters to correct LLMs' biases, but large $n$-virtual takes up more prompt space. We collect the accuracy and bias results of GPT-J-6B using ReBE under different $n$-virtual. According to standard deviation data in Table 4 and Figure 19 in the Appendix, there is no apparent relationship between $n$-virtual and bias.

**Order of Examples** Since LLMs are susceptible to position bias, previous work has found that the example order of a few-shot prompt affects the performance of ICL (Lu et al., 2022; Zhao et al., 2021). To reveal the effect of example order on ReBE, we shuffle the examples in the prompt under different random seeds. As shown in row "order" of Table 4 and Figure 19, the bias of LLM using ICL is not affected by the example order, and ReBE is also robust to changes in the example order.

# 6 RELATED WORK

After realizing that ICL performance is susceptible to example selection, many efforts have been made to stabilize it. Liu et al. (2022b) proposed the KATE, which retrieves examples semantically similar to the test query samples. After that, many heuristic-based methods have emerged, such as perplexity-based (Gonen et al., 2023; Iter et al., 2023), informativeness-based (Gupta et al., 2023; Li & Qiu, 2023) and sensitivity-based (Chen et al., 2023b). Besides that, some studies understand example selection from different perspectives, such as formulating it as a sequential decision problem (Zhang et al., 2022; Liu et al., 2024a), curating a stable subset from the original training set (Chang & Jia, 2023), selecting based on the Determinantal Point Process (DPP) (Yang et al., 2023; Ye et al., 2023) and Latent Variable Models (Wang et al., 2023). **Although these methods stabilize the accuracy of ICL on downstream tasks to a certain extent, they ignore the potential bias risks.** On the other hand, while extensive research has been conducted on the biases of LLMs (Gallegos et al., 2024), few studies focus on the bias risks of adapting LLMs to downstream tasks, especially for ICL. Although Ma et al. (2023) analyzed the predictive bias of ICL, their method relies on explicit bias attributes, making it inapplicable to the *EEC-paraphrase* dataset used in this paper. Additionally, predictive bias differs slightly from the social bias we focus on.

# 7 CONCLUSION

In this study, we have investigated the impact of example selection on the biases of LLMs. By comparing the biases under four example selection baselines with biases under zero-shot, we have found that example selection for ICL amplifies the biases of LLMs. To mitigate the bias of example selection, we have proposed the *Remind with Bias-aware Embedding* (ReBE), which removes the spurious correlations by contrastive learning and retains the feasibility of ICL by prompt tuning. After extensive experiments, we have demonstrated that ReBE can mitigate the bias without significantly compromising accuracy and is compatible with existing example selection methods. With the spread application of LLMs, more attention must be paid to the ethical risks of adapting LLMs to downstream tasks.

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

## A  DATASET

As shown in Table 6 and Table 7, compared to *EEC*, *EEC-paraphrase* built by this paper contains more complex and natural sentences. The sentences in *EEC-paraphrase* are obtained by constructing paraphrasing prompts in Table 6 and processing them with the help of `GPT-3.5-Turbo`.

Table 6: *EEC* and *EEC-paraphrase*.

| Dataset | Sentence example |
|---|---|
| *EEC* | "Alan feels angry." |
| *EEC-Paraphrase* | "Alan is experiencing a profound sense of frustration and irritation, resulting in a heightened state of emotional turmoil and discomfort." |
| Paraphrase Prompt | You will be give a sentence. Please paraphrase and expand the following sentence in more complex words (no less than 20 words; do not include 'joy', 'fear', 'sadness', and 'anger' in your answer.) without changing the original meaning: 
 {input} |

To compare the complexity and naturalness quantitatively, we conduct the following evaluation with the help of the Python library Textstat and provide results in Table 7. The performance of *EEC-paraphrase* on various metrics is significantly better than the original *EEC* dataset.

Table 7 shows the mean and standard deviation of the performance of datasets on various metrics. We choose the number of words and Distinct-n to measure the diversity of sentences, and metrics in Textstat are used to measure the complexity of sentences, which help determine the readability, complexity, and grade level.

Table 7: Evaluation of *EEC* and *EEC-paraphrase* on various metrics.

| | Metric | EEC | EEC-paraphrase |
|---|---|---|---|
| Diversity | Number of words↑ | 5.86(±1.73) | 18.63(±2.33) |
| | Distinct-2 ↑ | 0.81(±0.066) | 0.94(±0.147) |
| | Distinct-3 ↑ | 0.62(±0.132) | 0.89(±0.017) |
| Complexity | Automated Readability Index ↑ | 7.56(±3.80) | 14.44(±2.27) |
| | Coleman-Liau Index ↑ | 9.63(±4.57) | 14.74(±2.92) |
| | Dale-Chall Readability Score ↑ | 11.94(±3.21) | 11.68(±1.17) |
| | Flesch-Kincaid Grade Level ↑ | 5.52(±3.68) | 12.06(±2.04) |
| | Flesch Reading Ease Score ↓ | 65.88(±26.09) | 41.68(±14.38) |
| | Fog Scale ↑ | 8.44(±5.04) | 15.18(±2.88) |
| | Linsear Write Formula ↑ | 2.87(±1.25) | 12.83(±2.06) |
| | McAlpine EFLAW Readability Score ↑ | 7.34(±2.59) | 25.62(±3.90) |
| | Readability Consensus Score ↑ | 7.15(±4.45) | 13.09(±2.29) |
| | Spache Readability Formula ↑ | 4.20(±1.26) | 6.73(±0.70) |

## B  BIAS METRICS

In this section, we provide a detailed explanation of the bias metrics in Table 1.

**Average Group Fairness** (AvgGF) is a macro metric, which measures the disparity in the overall prediction accuracy between different groups. Formally, it is the absolute value of the difference between the accuracy $P(\hat{Y}=Y|S=s_1)$ of group $s_1$ and the accuracy $P(\hat{Y}=Y|S=s_2)$ of group $s_2$.

**Maximum TPR Gap** (MaxTG) is a metric derived from the True Positive Rate (TPR), which refers to the proportion of actual positive samples predicted to be positive. Here, we select one sentiment category as positive and the other three as negative. MaxTG measures the maximum recall (TPR) difference between various groups among all sentiment categories.

**Maximum FPR Gap** (MaxFG) is a metric derived from the False Positive Rate (FPR), which refers to the proportion of actual negative samples predicted to be positive. Here, we select one sentiment category as negative and the other three as positive. MaxFG measures the maximum FPR difference

between various groups among all sentiment categories. The larger the MaxFG, the samples from one group are more likely to be misclassified as a specific category than samples from another group. For example, the sentence *sadness* corresponding to *Male* in Figure 3 is more likely to be misclassified as *fear* than that corresponding to *Female*.

## C  IMPACTS OF EXAMPLE SELECTION ON LLMS BIASES

### C.1  IMPACTS OF THREE EXAMPLE SELECTION METHODS ON BIASES

In the main text, we provide the figure for the impacts of random-based example selection on biases of LLMs. Here, we provide results for three other example selection methods. As shown in Figure 8, Figure 9 and Figure 10, the maximum bias values of LLMs under ICL based on the three example selection methods are amplified to varying degrees. However, regarding the mean bias of LLMs, whether gender or race bias, all example selection methods except **perplexity-based** significantly reduce it.

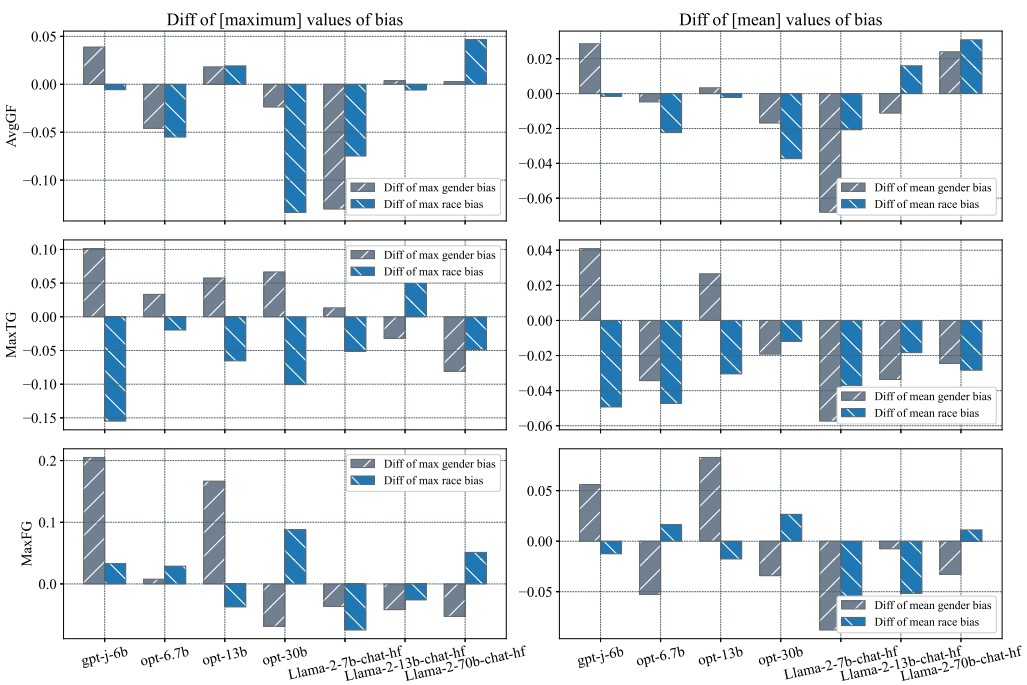

Figure 8: The impacts of **Perplexity-based** example selection on biases of LLMs.

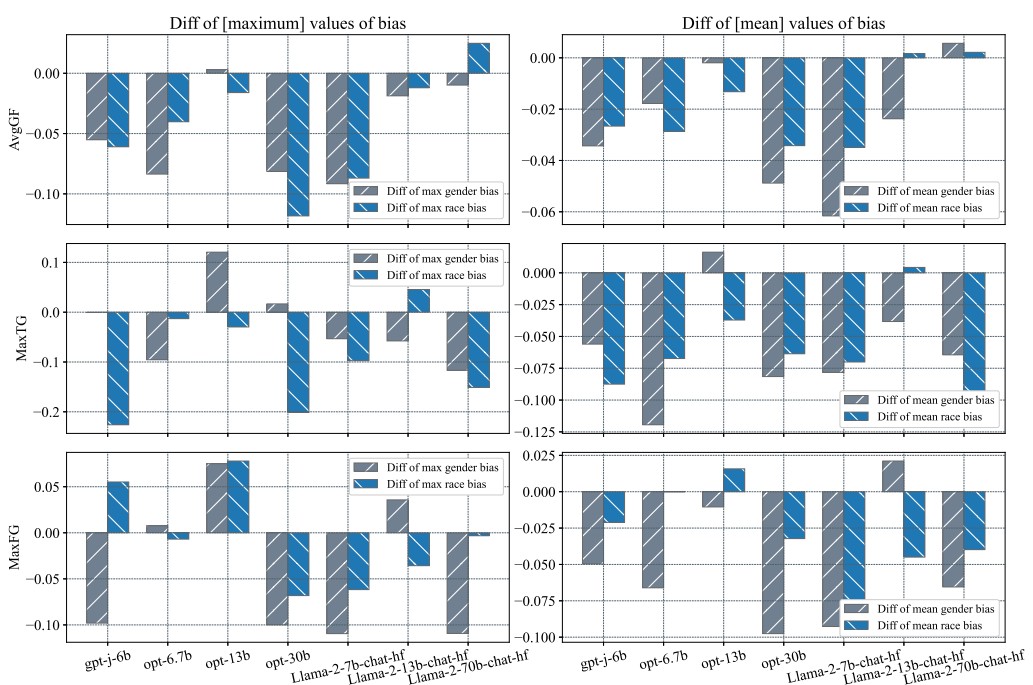

Figure 9: The impacts of **Similarity-based** example selection on biases of LLMs.

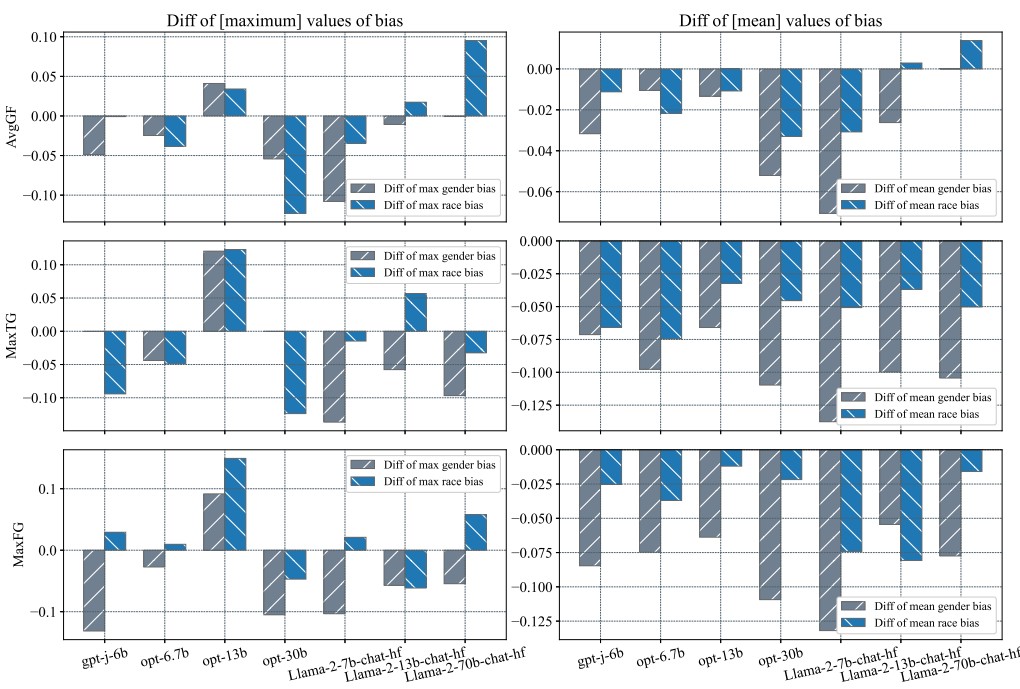

Figure 10: The impacts of **Dpp-based** example selection on biases of LLMs.

## C.2 Accuracy and **Race Bias** of LLMs under four example selection baselines

In addition to the gender bias in the main text, we collect the race bias data of eight LLMs under four example selection baselines. Detail experiment results of **race bias** of eight LLMs are shown in Table 8.

Table 8: Accuracy and **race bias** of LLMs under four example selection baselines.

| | | GPT-J-6B | GPT-neo-2.7B | OPT-6.7B | OPT-13B | OPT-30B | Llama-2-7B | Llama-2-13B | Llama-2-70B |
|---|---|---|---|---|---|---|---|---|---|
| **Random** | $\text{Acc}_{(Min)}$ | $0.84_{(0.80)}$ | $0.77_{(0.58)}$ | $0.81_{(0.67)}$ | $0.82_{(0.72)}$ | $0.84_{(0.76)}$ | $0.86_{(0.81)}$ | $0.87_{(0.83)}$ | $0.86_{(0.82)}$ |
| | $\text{AvgGF}_{(Max)}$ | $0.04_{(\mathbf{0.18})}$ | $0.05_{(0.15)}$ | $0.04_{(0.15)}$ | $0.05_{(0.11)}$ | $0.06_{(\mathbf{0.16})}$ | $0.06_{(0.12)}$ | $0.06_{(0.10)}$ | $0.08_{(0.12)}$ |
| | $\text{MaxTG}_{(Max)}$ | $0.15_{(\mathbf{0.32})}$ | $0.12_{(0.28)}$ | $0.13_{(0.29)}$ | $0.13_{(0.27)}$ | $0.14_{(\mathbf{0.30})}$ | $0.15_{(0.26)}$ | $0.16_{(0.22)}$ | $0.18_{(0.26)}$ |
| | $\text{MaxFG}_{(Max)}$ | $0.13_{(0.28)}$ | $0.12_{(0.24)}$ | $0.13_{(\mathbf{0.33})}$ | $0.14_{(0.26)}$ | $0.13_{(\mathbf{0.34})}$ | $0.13_{(0.23)}$ | $0.13_{(0.19)}$ | $0.16_{(0.23)}$ |
| **Perplexity** | $\text{Acc}_{(Min)}$ | $0.83_{(0.72)}$ | $0.79_{(0.61)}$ | $0.85_{(0.81)}$ | $0.83_{(0.79)}$ | $0.86_{(0.85)}$ | $0.87_{(0.80)}$ | $0.87_{(0.84)}$ | $0.86_{(0.85)}$ |
| | $\text{AvgGF}_{(Max)}$ | $0.05_{(0.09)}$ | $0.04_{(\mathbf{0.10})}$ | $0.03_{(0.07)}$ | $0.04_{(\mathbf{0.10})}$ | $0.03_{(0.07)}$ | $0.05_{(0.07)}$ | $0.05_{(0.06)}$ | $0.05_{(0.09)}$ |
| | $\text{MaxTG}_{(Max)}$ | $0.13_{(0.18)}$ | $0.13_{(0.26)}$ | $0.15_{(\mathbf{0.27})}$ | $0.12_{(0.15)}$ | $0.17_{(\mathbf{0.26})}$ | $0.12_{(0.18)}$ | $0.13_{(0.21)}$ | $0.15_{(0.24)}$ |
| | $\text{MaxFG}_{(Max)}$ | $0.12_{(0.18)}$ | $0.17_{(\mathbf{0.26})}$ | $0.18_{(0.25)}$ | $0.11_{(0.14)}$ | $0.17_{(\mathbf{0.34})}$ | $0.09_{(0.12)}$ | $0.13_{(0.21)}$ | $0.16_{(0.24)}$ |
| **Similarity** | $\text{Acc}_{(Min)}$ | $0.92_{(0.88)}$ | $0.85_{(0.82)}$ | $0.84_{(0.82)}$ | $0.87_{(0.86)}$ | $0.90_{(0.86)}$ | $0.93_{(0.90)}$ | $0.92_{(0.90)}$ | $0.89_{(0.87)}$ |
| | $\text{AvgGF}_{(Max)}$ | $0.02_{(0.04)}$ | $0.02_{(0.03)}$ | $0.03_{(\mathbf{0.08})}$ | $0.03_{(0.06)}$ | $0.04_{(\mathbf{0.08})}$ | $0.03_{(0.06)}$ | $0.03_{(0.06)}$ | $0.03_{(0.07)}$ |
| | $\text{MaxTG}_{(Max)}$ | $0.09_{(0.11)}$ | $0.11_{(\mathbf{0.24})}$ | $0.13_{(\mathbf{0.27})}$ | $0.11_{(0.19)}$ | $0.12_{(0.16)}$ | $0.09_{(0.13)}$ | $0.15_{(0.20)}$ | $0.08_{(0.14)}$ |
| | $\text{MaxFG}_{(Max)}$ | $0.11_{(0.20)}$ | $0.14_{(\mathbf{0.24})}$ | $0.16_{(0.21)}$ | $0.15_{(\mathbf{0.26})}$ | $0.11_{(0.18)}$ | $0.08_{(0.13)}$ | $0.14_{(0.20)}$ | $0.10_{(0.19)}$ |
| **DPP** | $\text{Acc}_{(Min)}$ | $0.93_{(0.89)}$ | $0.89_{(0.83)}$ | $0.87_{(0.79)}$ | $0.89_{(0.82)}$ | $0.91_{(0.86)}$ | $0.94_{(0.90)}$ | $0.93_{(0.91)}$ | $0.90_{(0.85)}$ |
| | $\text{AvgGF}_{(Max)}$ | $0.04_{(0.10)}$ | $0.04_{(\mathbf{0.16})}$ | $0.03_{(0.08)}$ | $0.03_{(0.11)}$ | $0.04_{(0.08)}$ | $0.04_{(0.11)}$ | $0.04_{(0.09)}$ | $0.04_{(\mathbf{0.14})}$ |
| | $\text{MaxTG}_{(Max)}$ | $0.11_{(0.24)}$ | $0.13_{(\mathbf{0.34})}$ | $0.12_{(0.24)}$ | $0.12_{(\mathbf{0.34})}$ | $0.14_{(0.23)}$ | $0.11_{(0.22)}$ | $0.11_{(0.22)}$ | $0.13_{(0.26)}$ |
| | $\text{MaxFG}_{(Max)}$ | $0.10_{(0.18)}$ | $0.12_{(\mathbf{0.35})}$ | $0.13_{(0.23)}$ | $0.12_{(\mathbf{0.33})}$ | $0.12_{(0.20)}$ | $0.10_{(0.22)}$ | $0.10_{(0.18)}$ | $0.13_{(0.25)}$ |

$\text{Avg}_{(Min)}$ are the largest two values in AvgGF; $\text{Avg}_{Min}$ are the largest two values in MaxTG and MaxFG.

## C.3 Confusion Matrices

To prove the existence of spurious correlations, we provide all the confusion matrices of eight LLMs in 11 and Figure 12. It can be seen that, except for `OPT-13B`, all LLMs exhibit spurious correlations, which is consistent with the results in Table 8.

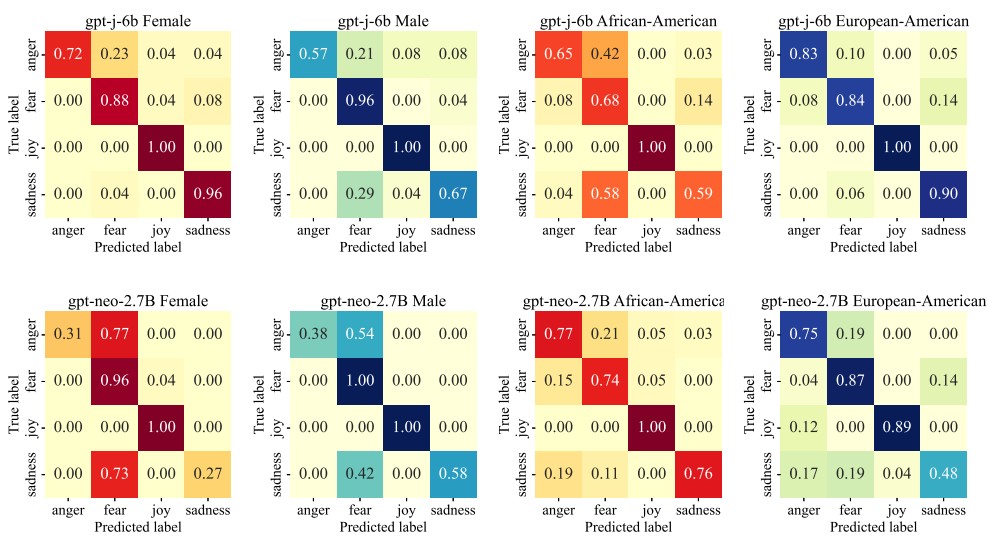

Figure 11: Confusion matrix heatmaps.

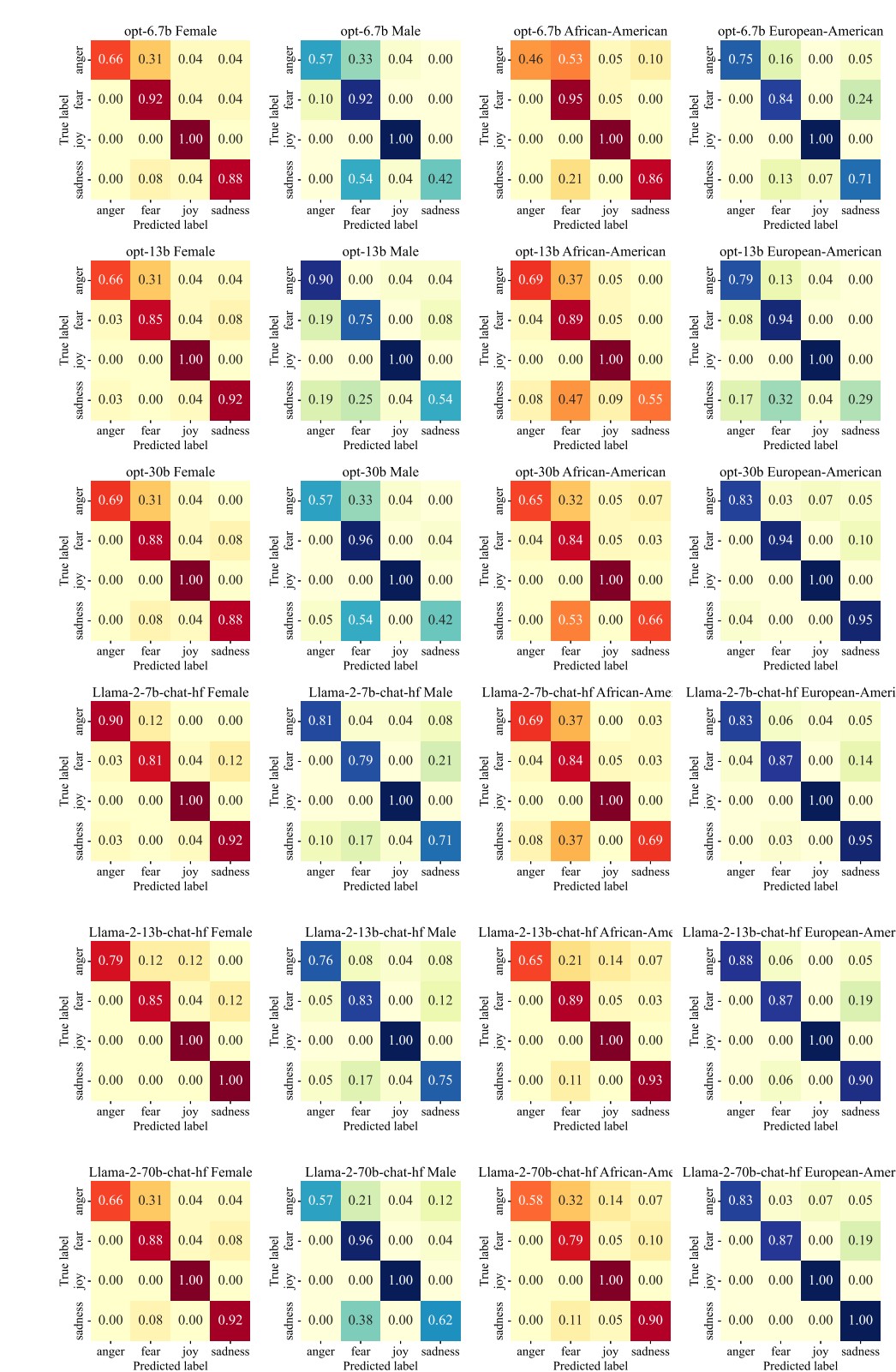

Figure 12: Confusion matrix heatmaps (continuation).

## C.4 THE NATIVE **RACE BIAS** OF LLMS

Native Race Bias of LLMs are shown in Figure 13.

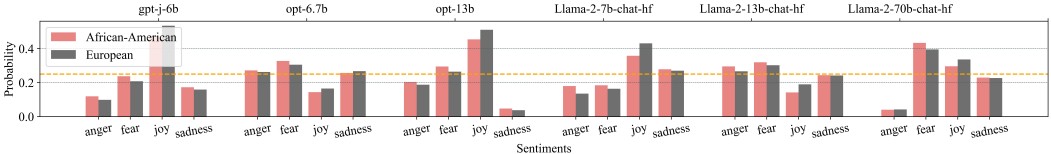

Figure 13: The native race bias of LLMs over different labels.

## C.5 BIAS DISTRIBUTION

Feature distribution of ICL-prompts with high and low MaxFG is shown in Figure 14.

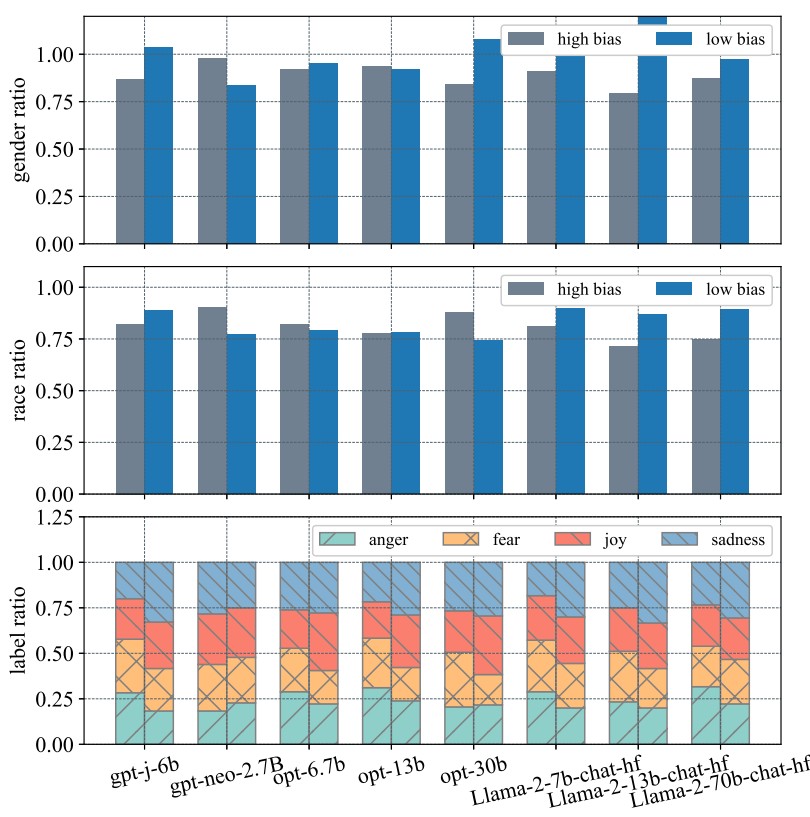

Figure 14: The feature distributions of few-shot prompts with high and low MaxFG bias.

# D   EXPERIMENTAL RESULTS AFTER DEBIASING BY ReBE

## D.1   GENDER BIAS RESULTS AFTER DEBIASING BY ReBE

To supplement the data in Table 3, we provide the accuracy and gender bias comparison of `OPT-6.7B` (Figure 15) and `OPT-13B` (Figure 17). Furthermore, for comparison, we also provide the distribution of race bias in Figure 16 and Figure 18.

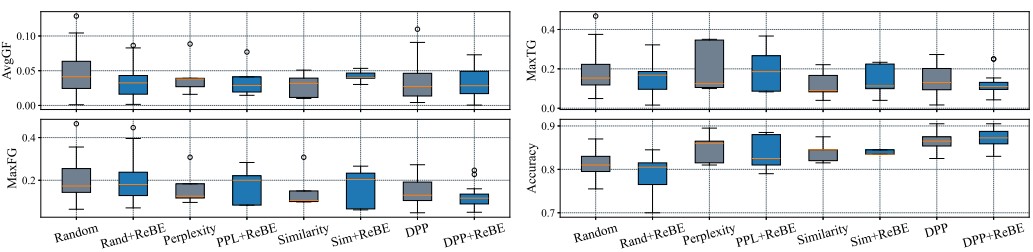

Figure 15: The accuracy and gender bias comparison of `OPT-6.7B` under four example selection baselines before and after debias.

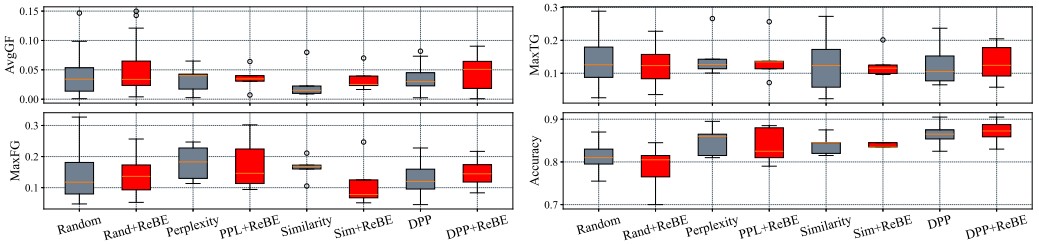

Figure 16: The accuracy and race bias comparison of `OPT-6.7B` under four example selection baselines before and after debias.

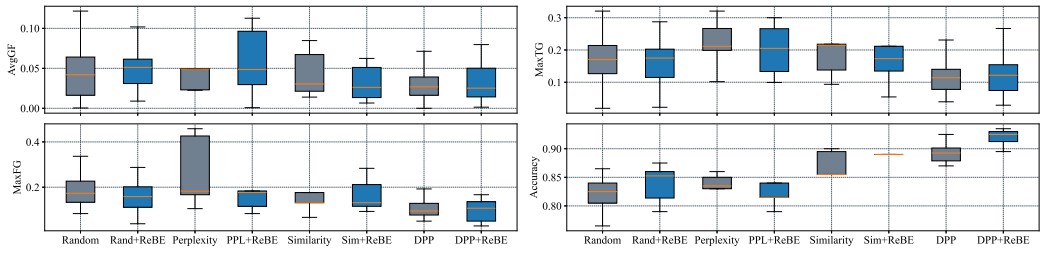

Figure 17: The accuracy and gender bias comparison of `OPT-13B` under four example selection baselines before and after debias.

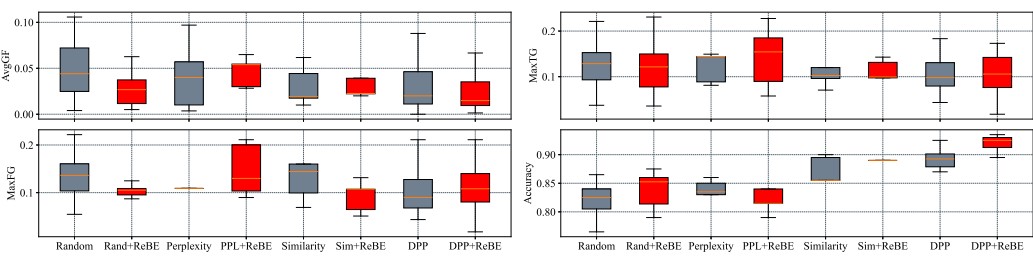

Figure 18: The accuracy and race bias comparison of OPT-13B under four example selection baselines before and after debias.

## D.2 RACE BIAS RESULTS AFTER DEBIASING BY ReBE

Similar to the gender bias, according to the results in Table 8, we select the LLM with the largest bias in each baseline to eliminate the race bias. The results are shown in Table 9.

Table 9: **Race** bias and accuracy of LLMs under example selections after debiasing.

|  |  | Acc↑ | AvgGF↓ | MaxTG↓ | MaxFG↓ |
|---|---|---|---|---|---|
| **Random** | Max | **GPT-J-6B** | $0.140_{(-0.035)}$ | $0.256_{(-0.062)}$ | $0.228_{(-0.057)}$ |
|  | Avg | $0.843_{(+0.000)}$ | $0.037_{(-0.006)}$ | $0.148_{(+0.002)}$ | $0.132_{(+0.006)}$ |
| **Perplexity** | Max | **GPT-neo-2.7B** | $0.076_{(-0.018)}$ | $0.195_{(-0.062)}$ | $0.211_{(-0.045)}$ |
|  | Avg | $0.807_{(+0.017)}$ | $0.041_{(-0.000)}$ | $0.121_{(-0.007)}$ | $0.128_{(-0.038)}$ |
| **Similarity** | Max | **OPT-6.7B** | $0.065_{(-0.015)}$ | $0.147_{(-0.125)}$ | $0.224_{(+0.013)}$ |
|  | Avg | $0.862_{(+0.022)}$ | $0.030_{(+0.003)}$ | $0.108_{(-0.021)}$ | $0.137_{(-0.027)}$ |
| **DPP** | Max | **GPT-neo-2.7B** | $0.150_{(-0.015)}$ | $0.264_{(-0.081)}$ | $0.271_{(-0.074)}$ |
|  | Avg | $0.883_{(-0.004)}$ | $0.040_{(+0.001)}$ | $0.140_{(+0.014)}$ | $0.144_{(+0.024)}$ |

[1] Red subscript indicates that the metric increases after debiasing, and blue subscript indicates that the metric decreases after debiasing.

## D.3 DETAILED EXPERIMENTAL RESULTS OF PARAMETER ANALYSIS

In this subsection, we provide detailed data on the effect of example order, $k$-shot and $n$-virtual on ReBE in Table 10, Table 11 and Table 12 respectively.

Table 10: Detailed data on the effect of example order on ReBE.

| Shuffle Seed | Accuracy↑ | | AvgGF↓ | | MaxTG↓ | | MaxFG↓ | |
|---|---|---|---|---|---|---|---|---|
|  | Mean | Min | Mean | Max | Mean | Max | Mean | Max |
| 13 | 0.826 | 0.730 | 0.034 | 0.074 | 0.140 | 0.276 | 0.181 | 0.332 |
| 21 | 0.829 | 0.755 | 0.031 | 0.073 | 0.143 | 0.319 | 0.188 | 0.379 |
| 42 | 0.837 | 0.775 | 0.030 | 0.087 | 0.131 | 0.244 | 0.171 | 0.284 |
| 87 | 0.828 | 0.665 | 0.032 | 0.092 | 0.128 | 0.295 | 0.166 | 0.279 |
| 100 | 0.837 | 0.790 | 0.030 | 0.126 | 0.128 | 0.234 | 0.157 | 0.263 |

## E IMPACT OF THE NUMBER OF ICL EXAMPLES ON BIAS

To investigate the impact of increasing the number of IC examples, we have assessed the gender bias performance of Llama-2-7B in toxicity detection under various number of ICL examples ($k \in [2, 6, 10, 14, 18, 22, 26]$). As the number of ICL examples $k$ increases, the bias decreases

Table 11: Detailed data on the effect of $k$-shot on ReBE.

| $k$-shot | Accuracy↑ | | AvgGF↓ | | MaxTG↓ | | MaxFG↓ | |
|---|---|---|---|---|---|---|---|---|
| | Mean | Min | Mean | Max | Mean | Max | Mean | Max |
| 2 | 0.747 | 0.540 | 0.042 | 0.147 | 0.156 | 0.288 | 0.157 | 0.288 |
| 6 | 0.787 | 0.670 | 0.044 | 0.121 | 0.156 | 0.304 | 0.175 | 0.374 |
| 10 | 0.800 | 0.650 | 0.038 | 0.111 | 0.148 | 0.253 | 0.190 | 0.355 |
| 14 | 0.806 | 0.610 | 0.033 | 0.095 | 0.141 | 0.308 | 0.195 | 0.314 |
| 18 | 0.837 | 0.775 | 0.033 | 0.082 | 0.136 | 0.221 | 0.179 | 0.284 |
| 22 | 0.807 | 0.685 | 0.032 | 0.095 | 0.141 | 0.396 | 0.174 | 0.443 |
| 26 | 0.823 | 0.760 | 0.031 | 0.147 | 0.141 | 0.401 | 0.191 | 0.407 |

Table 12: Detailed data on the effect of $n$-virtual on ReBE.

| $n$-virtual | Accuracy↑ | | AvgGF↓ | | MaxTG↓ | | MaxFG↓ | |
|---|---|---|---|---|---|---|---|---|
| | Mean | Min | Mean | Max | Mean | Max | Mean | Max |
| 1 | 0.853 | 0.825 | 0.033 | 0.083 | 0.135 | 0.301 | 0.145 | 0.253 |
| 3 | 0.850 | 0.795 | 0.030 | 0.073 | 0.143 | 0.292 | 0.159 | 0.263 |
| 5 | 0.843 | 0.790 | 0.027 | 0.111 | 0.134 | 0.301 | 0.160 | 0.346 |
| 10 | 0.837 | 0.775 | 0.033 | 0.082 | 0.136 | 0.221 | 0.179 | 0.284 |
| 20 | 0.866 | 0.820 | 0.043 | 0.088 | 0.148 | 0.250 | 0.151 | 0.256 |
| 30 | 0.856 | 0.820 | 0.034 | 0.092 | 0.137 | 0.250 | 0.158 | 0.250 |
| 50 | 0.832 | 0.800 | 0.018 | 0.049 | 0.070 | 0.135 | 0.108 | 0.148 |
| 100 | 0.828 | 0.780 | 0.045 | 0.113 | 0.146 | 0.288 | 0.128 | 0.215 |

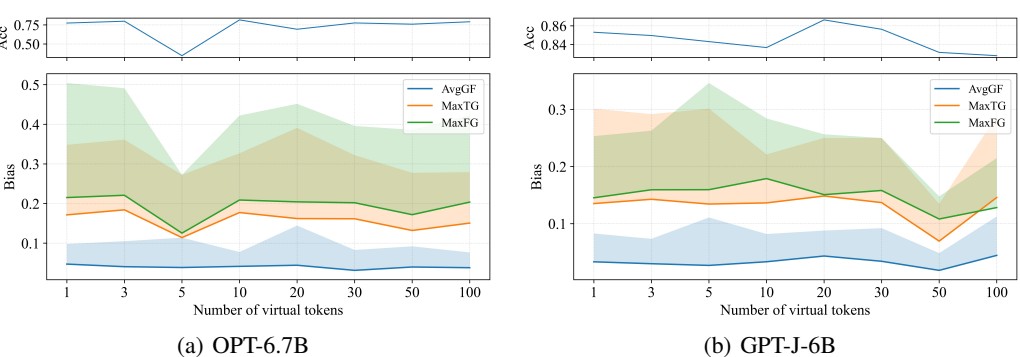

(a) OPT-6.7B  (b) GPT-J-6B

Figure 19: The effect of $n$-virtual on ReBE.

overall, but the change in accuracy must also be considered. The results are presented in Table 13, Table 14, Table 15 and Table 16.

Table 13: Bias performance of `Llama-2-7B` on $AvgGF$

| | Random | | Perplexity | | Similarity | | DPP | |
|---|---|---|---|---|---|---|---|---|
| | Mean | Max | Mean | Max | Mean | Max | Mean | Max |
| $k=2$ | 0.204($\pm$0.05) | 0.310 | 0.105($\pm$0.05) | 0.195 | 0.116($\pm$0.05) | 0.203 | 0.129($\pm$0.05) | 0.235 |
| $k=6$ | 0.204($\pm$0.05) | 0.312 | 0.087($\pm$0.06) | 0.211 | 0.067($\pm$0.04) | 0.166 | 0.073($\pm$0.05) | 0.195 |
| $k=10$ | 0.175($\pm$0.03) | 0.247 | 0.075($\pm$0.06) | 0.187 | 0.046($\pm$0.03) | 0.158 | 0.062($\pm$0.04) | 0.156 |
| $k=14$ | 0.179($\pm$0.05) | 0.263 | 0.083($\pm$0.05) | 0.189 | 0.041($\pm$0.03) | 0.127 | 0.056($\pm$0.03) | 0.105 |
| $k=18$ | 0.179($\pm$0.05) | 0.283 | 0.058($\pm$0.06) | 0.205 | 0.043($\pm$0.05) | 0.154 | 0.051($\pm$0.04) | 0.136 |
| $k=22$ | 0.187($\pm$0.05) | 0.285 | 0.064($\pm$0.06) | 0.211 | 0.035($\pm$0.03) | 0.109 | 0.041($\pm$0.03) | 0.094 |
| $k=26$ | 0.151($\pm$0.04) | 0.229 | 0.063($\pm$0.05) | 0.198 | 0.038($\pm$0.03) | 0.130 | 0.046($\pm$0.02) | 0.078 |

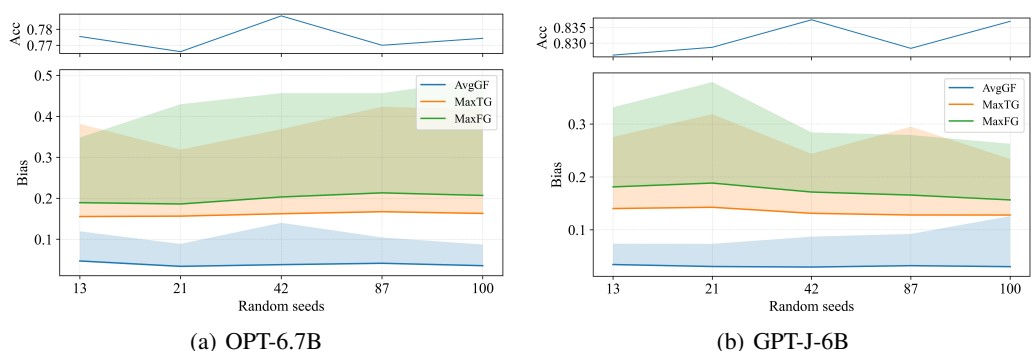

(a) OPT-6.7B  (b) GPT-J-6B

Figure 20: The effect of example order on ReBE.

Table 14: Bias performance of `Llama-2-7B` on $MaxTG$

| | Random | | Perplexity | | Similarity | | DPP | |
|---|---|---|---|---|---|---|---|---|
| | Mean | Max | Mean | Max | Mean | Max | Mean | Max |
| $k=2$ | 0.220($\pm$0.05) | 0.322 | 0.114($\pm$0.05) | 0.203 | 0.123($\pm$0.06) | 0.223 | 0.139($\pm$0.06) | 0.251 |
| $k=6$ | 0.226($\pm$0.05) | 0.336 | 0.094($\pm$0.06) | 0.236 | 0.067($\pm$0.04) | 0.166 | 0.077($\pm$0.06) | 0.210 |
| $k=10$ | 0.192($\pm$0.03) | 0.261 | 0.081($\pm$0.06) | 0.207 | 0.050($\pm$0.03) | 0.151 | 0.066($\pm$0.04) | 0.160 |
| $k=14$ | 0.196($\pm$0.05) | 0.278 | 0.089($\pm$0.06) | 0.194 | 0.042($\pm$0.03) | 0.109 | 0.062($\pm$0.03) | 0.114 |
| $k=18$ | 0.195($\pm$0.05) | 0.312 | 0.065($\pm$0.06) | 0.217 | 0.044($\pm$0.03) | 0.140 | 0.059($\pm$0.04) | 0.156 |
| $k=22$ | 0.204($\pm$0.05) | 0.305 | 0.075($\pm$0.06) | 0.228 | 0.036($\pm$0.03) | 0.103 | 0.045($\pm$0.03) | 0.107 |
| $k=26$ | 0.162($\pm$0.05) | 0.254 | 0.071($\pm$0.06) | 0.215 | 0.039($\pm$0.03) | 0.129 | 0.049($\pm$0.05) | 0.094 |

Table 15: Bias performance of `Llama-2-7B` on $MaxFG$

| | Random | | Perplexity | | Similarity | | DPP | |
|---|---|---|---|---|---|---|---|---|
| | Mean | Max | Mean | Max | Mean | Max | Mean | Max |
| $k=2$ | 0.025($\pm$0.07) | 0.250 | 0.069($\pm$0.09) | 0.2500 | 0.123($\pm$0.18) | 0.667 | 0.180($\pm$0.15) | 0.500 |
| $k=6$ | 0.168($\pm$0.12) | 0.250 | 0.198($\pm$0.17) | 0.500 | 0.091($\pm$0.14) | 0.500 | 0.104($\pm$0.15) | 0.500 |
| $k=10$ | 0.135($\pm$0.12) | 0.250 | 0.182($\pm$0.16) | 0.400 | 0.079($\pm$0.14) | 0.500 | 0.120($\pm$0.15) | 0.500 |
| $k=14$ | 0.133($\pm$0.12) | 0.300 | 0.199($\pm$0.15) | 0.500 | 0.108($\pm$0.17) | 0.500 | 0.142($\pm$0.17) | 0.500 |
| $k=18$ | 0.104($\pm$0.11) | 0.250 | 0.182($\pm$0.18) | 0.667 | 0.126($\pm$0.17) | 0.500 | 0.171($\pm$0.18) | 0.667 |
| $k=22$ | 0.128($\pm$0.14) | 0.500 | 0.285($\pm$0.20) | 0.667 | 0.113($\pm$0.14) | 0.500 | 0.154($\pm$0.16) | 0.500 |
| $k=26$ | 0.095($\pm$0.11) | 0.300 | 0.306($\pm$0.20) | 0.667 | 0.104($\pm$0.14) | 0.500 | 0.176($\pm$0.15) | 0.500 |

Table 16: Accuracy of `Llama-2-7B` under vairous $k$

| | Random | Perplexity | Similarity | DPP |
|---|---|---|---|---|
| $k=2$ | 0.721($\pm$0.08) | 0.669($\pm$0.12) | 0.676($\pm$0.02) | 0.688($\pm$0.03) |
| $k=6$ | 0.744($\pm$0.07) | 0.715($\pm$0.13) | 0.716($\pm$0.04) | 0.786($\pm$0.03) |
| $k=10$ | 0.757($\pm$0.07) | 0.803($\pm$0.09) | 0.765($\pm$0.03) | 0.824($\pm$0.03) |
| $k=14$ | 0.776($\pm$0.07) | 0.817($\pm$0.06) | 0.778($\pm$0.03) | 0.837($\pm$0.03) |
| $k=18$ | 0.762($\pm$0.06) | 0.830($\pm$0.08) | 0.802($\pm$0.03) | 0.850($\pm$0.03) |
| $k=22$ | 0.769($\pm$0.06) | 0.839($\pm$0.06) | 0.827($\pm$0.04) | 0.880($\pm$0.03) |
| $k=26$ | 0.805($\pm$0.06) | 0.869($\pm$0.04) | 0.851($\pm$0.02) | 0.889($\pm$0.03) |

# F  EXPERIMENTS OF TOXICITY DETECTION

In toxicity detection, LLMs are asked to judge whether the sentences given are toxic or non-toxic. As shown in Table 17 and Table 18, we provide the gender bias performance of `Llama-2-7B` and `Llama-3.2-3B` in toxicity detection.

Consistent with the sentiment analysis, we can find that: ❶ Compared with the zero-shot, example selection methods for ICL amplify the maximum value of gender bias; ❷ ReBE remains compatible with example selection methods and exhibits effective debiasing in toxicity detection.

Table 17: Maximum values of gender bias of `Llama-2-7B` in toxicity dection

| Llama-2-7B | $AvgGF$ | | $MaxTG$ | | $MaxFG$ | |
|---|---|---|---|---|---|---|
| $k = 18$ | Origin | $ReBE$ | Origin | $ReBE$ | Origin | $ReBE$ |
| Zero-shot | 0.108 | - | 0.098 | - | 0.833 | - |
| Random-based | 0.283 | 0.186 | 0.312 | 0.210 | 0.250 | 0.300 |
| Perplexity-based | 0.205 | 0.168 | 0.217 | 0.173 | 0.667 | 0.667 |
| Similarity-based | 0.154 | 0.141 | 0.140 | 0.129 | 0.500 | 0.667 |
| DPP-based | 0.136 | 0.102 | 0.156 | 0.116 | 0.667 | 0.857 |

Table 18: Maximum values of gender bias of `Llama-3.2-3B` in toxicity dection

| Llama-3.2-3B | $AvgGF$ | | $MaxTG$ | | $MaxFG$ | |
|---|---|---|---|---|---|---|
| $k = 18$ | Origin | $ReBE$ | Origin | $ReBE$ | Origin | $ReBE$ |
| Zero-shot | 0.145 | - | 0.158 | - | 0.429 | - |
| Random-based | 0.215 | 0.108 | 0.217 | 0.127 | 0.500 | 0.550 |
| Perplexity-based | 0.142 | 0.043 | 0.152 | 0.019 | 0.857 | 0.333 |
| Similarity-based | 0.056 | 0.038 | 0.069 | 0.019 | 0.600 | 0.333 |
| DPP-based | 0.090 | 0.048 | 0.049 | 0.011 | 0.750 | 0.500 |

Table 19: Mean values of gender bias of `Llama-2-7B` in toxicity dection

| Llama-2-7B | AvgGF | | MaxTG | | MaxFG | |
|---|---|---|---|---|---|---|
| k=18 | Origin | ReBE | Origin | ReBE | Origin | ReBE |
| Zero-shot | 0.043(±0.03) | - | 0.053(±0.03) | - | 0.267(±0.25) | - |
| Random-based | 0.179(±0.05) | 0.058(±0.04) | 0.195(±0.05) | 0.070(±0.04) | 0.104(±0.11) | 0.176(±0.11) |
| Perplexity-based | 0.058(±0.06) | 0.049(±0.04) | 0.065(±0.06) | 0.053(±0.05) | 0.180(±0.18) | 0.256(±0.03) |
| Similarity-based | 0.043(±0.03) | 0.038(±0.03) | 0.044(±0.03) | 0.039(±0.03) | 0.126(±0.17) | 0.155(±0.20) |
| DPP-based | 0.051(±0.04) | 0.045(±0.03) | 0.059(±0.16) | 0.053(±0.02) | 0.171(±0.18) | 0.248(±0.20) |

Table 20: Mean values of gender bias of `Llama-3.2-3B` in toxicity dection

| Llama-3.2-3B | AvgGF | | MaxTG | | MaxFG | |
|---|---|---|---|---|---|---|
| k=18 | Origin | ReBE | Origin | ReBE | Origin | ReBE |
| Zero-shot | 0.057(±0.05) | - | 0.059(±0.05) | - | 0.177(±0.15) | - |
| Random-based | 0.058(±0.05) | 0.038(±0.03) | 0.056(±0.05) | 0.044(±0.03) | 0.126(±0.14) | 0.152(±0.11) |
| Perplexity-based | 0.038(±0.04) | 0.027(±0.01) | 0.038(±0.04) | 0.002(±0.01) | 0.263(±0.22) | 0.045(±0.10) |
| Similarity-based | 0.031(±0.06) | 0.024(±0.02) | 0.034(±0.02) | 0.006(±0.01) | 0.223(±0.19) | 0.150(±0.14) |
| DPP-based | 0.028(±0.03) | 0.03(±0.01) | 0.021(±0.02) | 0.004(±0.01) | 0.301(±0.23) | 0.070(±0.14) |

# G    BASELINE COMPARISON OF REBE

According to our survey results, there are not many relative debiasing methods of ICL. Although Hu et al. (2024) proposed fairness via clustering genetic (FCG) algorithm, it cannot apply to more tasks due to the need for explicit feature vectors. Therefore, we cannot set FCG as the baseline of ReBE. Since there are no other debiasing methods specifically for ICL, we compare ReBE with two context augmentation methods: counterfactual context and gender-balanced context.

**Counterfactual**    For datasets built based on templates, such as *EEC* and *EEC-paraphrase*, we can construct the corresponding counterfactual context instance based on the template. For example, according to the template `<person subject> feels <emotion word>`, the counterfactual context instance of sentence `Alonzo feels angry.` can be `Nichelle feels angry.` or `Amanda feels angry.`

**Gender-balanced**    The gender-balanced context approach requires an equal or close number of examples for each gender type.

Table 21 shows the gender bias of `OPT-6.7B` on sentiment analysis with *EEC-paraphrase* as the dataset. While the *EEC-paraphrase* dataset does not have its own templates, it is built upon the *EEC* samples, allowing us to generate counterfactual samples using the templates from the *EEC*.

Since the counterfactual context method does not apply to datasets without templates, we remove it from the baselines tested on the *Jigsaw* dataset. Table 22 shows the gender bias of `Llama-2-7B` on toxicity detection with *Jigsaw* as the dataset.

Table 21: Gender bias of `OPT-6.7B` on Sentiment Analysis with *EEC-paraphrase*

|  | $AvgGF$ (Mean) | Max | $MaxTG$ (Mean) | Max | $MaxFG$ (Mean) | Max | Acc |
|---|---|---|---|---|---|---|---|
| Random | 0.044($\pm$0.03) | 0.129 | 0.180($\pm$0.09) | 0.468 | 0.199($\pm$0.09) | 0.465 | 0.81 |
| DPP | 0.036($\pm$0.03) | 0.110 | 0.142($\pm$0.08) | 0.273 | 0.144($\pm$0.06) | 0.273 | 0.87 |
| Gender-balanced | 0.040($\pm$0.03) | 0.132 | 0.174($\pm$0.08) | 0.333 | 0.210($\pm$0.09) | 0.417 | 0.80 |
| Counterfactual | 0.035($\pm$0.03) | 0.125 | 0.145($\pm$0.07) | 0.369 | 0.149($\pm$0.07) | 0.369 | 0.77 |
| Random+ReBE | 0.034($\pm$0.02) | 0.086 | 0.151($\pm$0.07) | 0.322 | 0.191($\pm$0.08) | 0.447 | 0.78 |
| DPP+ReBE | 0.033($\pm$0.02) | 0.073 | 0.120($\pm$0.05) | 0.250 | 0.122($\pm$0.05) | 0.247 | 0.87 |

Table 22: Gender bias of `Llam-2-7B` on Toxicity detection with *Jigsaw*

|  | $AvgGF$ (Mean) | Max | $MaxTG$ (Mean) | Max | $MaxFG$ (Mean) | Max | Acc |
|---|---|---|---|---|---|---|---|
| Random | 0.179($\pm$0.05) | 0.283 | 0.215($\pm$0.05) | 0.312 | 0.215($\pm$0.05) | 0.312 | 0.76 |
| DPP | 0.051($\pm$0.04) | 0.136 | 0.059($\pm$0.04) | 0.156 | 0.171($\pm$0.18) | 0.667 | 0.85 |
| Gender-balanced | 0.116($\pm$0.06) | 0.236 | 0.205($\pm$0.08) | 0.500 | 0.205($\pm$0.08) | 0.500 | 0.81 |
| Random+ReBE | 0.058($\pm$0.04) | 0.186 | 0.070($\pm$0.04) | 0.210 | 0.176($\pm$0.11) | 0.300 | 0.86 |
| DPP+ReBE | 0.045($\pm$0.03) | 0.102 | 0.053($\pm$0.02) | 0.116 | 0.248($\pm$0.20) | 0.857 | 0.88 |

## G.1 DISCUSSION ON DEBIASING METHODS

To avoid spurious correlations caused by example selection, we test the feasibility of calibrating the prompts of ICL. Based on past experiences, we provide detailed instructions or construct the counterfactual example pairs in the prompt. Considering that differences in the proportion of sentences corresponding to demographic groups or labels might mislead the LLMs, we also try the approach that balances the proportions of various features in the prompt. However, as shown in Figure 21, there are no significant differences in feature ratios between prompts with high and low bias, so the method based on balanced features may not be effective enough. In sum, the feasibility of removing spurious correlations by few-shot prompts alone is questionable.

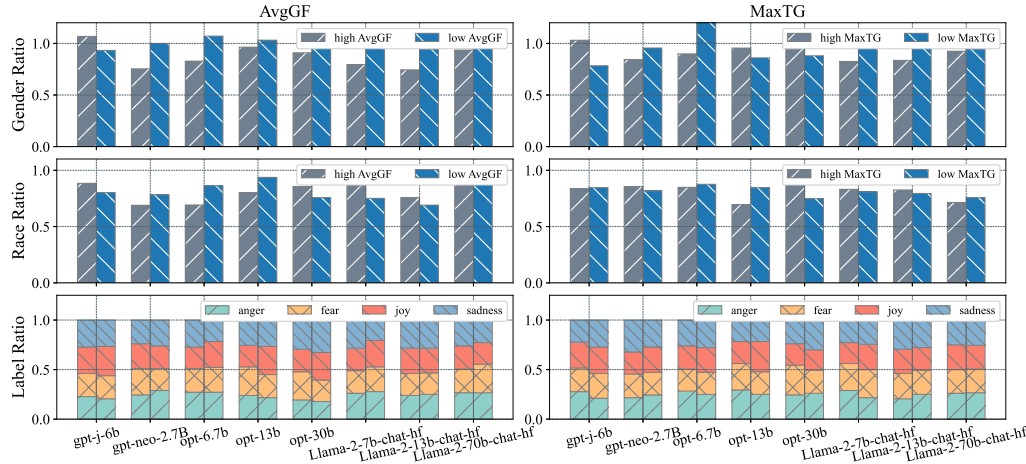

Figure 21: The feature distributions of ICL-prompts with high and low bias, which are constructed by Random-based example selection.

