# OpenReview forum: "Does Example Selection for In-Context Learning Amplify the Biases of Large Language Models?"
_ICLR.cc/2025/Conference — Submitted to ICLR 2025_

### Official Review · Reviewer_Av69 · 2024-10-30

**Soundness:** 3
**Presentation:** 3
**Contribution:** 3
**Rating:** 5
**Confidence:** 2

**Summary:**

This paper examines how example selection in in-context learning (ICL) can amplify biases in Large Language Models (LLMs). The authors make three key discoveries: example selection with high accuracy doesn't guarantee low bias, ICL example selection amplifies existing LLM biases, and it contributes to spurious correlations. To study these effects, they create EEC-paraphrase, a new sentiment classification dataset, and propose ReBE (Remind with Bias-aware Embedding), a novel debiasing method that combines contrastive learning with prompt tuning. Their results show that ReBE effectively reduces bias without significantly compromising accuracy while remaining compatible with existing example selection methods.

**Strengths:**

1. The paper identifies and investigates a new problem: how example selection in ICL affects model bias, an important angle that previous work has overlooked.

2. They creates a new dataset (EEC-paraphrase) that improves upon existing bias evaluation datasets by using more natural language and also proposes an innovative solution (ReBE) that creatively combines contrastive learning with prompt tuning.

3. . The authors test their findings across 8 different LLMs of varying sizes and 4 distinct example selection methods, using multiple bias metrics (AvgGF, MaxTG, MaxFG) to ensure comprehensive assessment. The authors also provide solid theoretical grounding to explain why their approach is effective.

**Weaknesses:**

1. A major methodological weakness lies in the dataset construction. Using GPT-3.5-Turbo to create EEC-paraphrase raises questions about potential inherited biases and quality control. The paper lacks discussion of human evaluation or validation of the paraphrased outputs.

2. The experimental evaluation would benefit from broader comparisons. The absence of comparisons with existing debiasing methods and fine-tuning approaches makes it difficult to fully assess ReBE's advantages. Including baselines like data augmentation or adversarial debiasing would provide valuable context.

**Questions:**

1. I'm concerned about potential biases introduced by using GPT-3.5-Turbo to create EEC-paraphrase. Could you describe your validation process and any controls implemented to ensure dataset quality? Human evaluation results would be particularly valuable.

2. The paper's findings about bias amplification are interesting but limited to sentiment classification. Have you explored whether these patterns hold true for other tasks like toxicity detection or text classification? Even preliminary results would help assess generalizability.

---

> ### Author Response · Authors · 2024-11-24
> **Author Response - Part 1**
>
> We appreciate the reviewer’s recognition of the significance of our work. Thank you for your valuable and thoughtful comments. Please find our point-by-point responses below:
>
> **Weakness 1 (Validation of *EEC-paraphrase*)**
>
> > **A major methodological weakness lies in the dataset construction. Using GPT-3.5-Turbo to create EEC-paraphrase raises questions about potential inherited biases and quality control.**
>
> Thank you for your valuable feedback. First, for quality control, **`previous work has demonstrated the capability of LLMs (including GPT-3.5-Turbo) to produce diverse and valid paraphrases under guidance`** [1]. We further **validate the quality** of *EEC-paraphrase* with the help of the Python library [Textstat](https://textstat.org/) and provide results in the table below.
>
> On the other hand, **`we have introduced a new task - toxicity detection, and believe that multiple tasks and multiple LLMs can minimize the impact of these potential biases on the results. We have also sampled and manually checked the generated samples.`**
>
> The table below shows the mean and standard deviation of the performance of datasets on various metrics. We choose the number of words and Distinct-n to measure the diversity of sentences, and metrics in [Textstat](https://textstat.org/) are used to measure the complexity of sentences, which help determine the readability, complexity, and grade level.
>
> || Metric| EEC| EEC-paraphrase|
> | - | -- | - | - |
> | Diversity  | Number of words$\uparrow$  | 5.86($\pm$1.73)| **18.63($\pm$2.33)**  |
> || Distinct-2 $\uparrow$| 0.81($\pm$0.066)  | **0.94($\pm$0.147)**  |
> || Distinct-3 $\uparrow$| 0.62($\pm$0.132)  | **0.89($\pm$0.017)**  |
> | Complexity | Automated Readability Index $\uparrow$| 7.56($\pm$3.80)| **14.44($\pm$2.27)**  |
> || Coleman-Liau Index $\uparrow$| 9.63($\pm$4.57)| **14.74($\pm$2.92)**  |
> || Dale-Chall Readability Score $\uparrow$| 11.94($\pm$3.21)  | **11.68($\pm$1.17)**  |
> || Flesch-Kincaid Grade Level $\uparrow$| 5.52($\pm$3.68)| **12.06($\pm$2.04)**  |
> || Flesch Reading Ease Score $\downarrow$| 65.88($\pm$26.09) | **41.68($\pm$14.38)** |
> || Fog Scale $\uparrow$| 8.44($\pm$5.04)| **15.18($\pm$2.88)**  |
> || Linsear Write Formula $\uparrow$| 2.87($\pm$1.25)| **12.83($\pm$2.06)**  |
> || McAlpine EFLAW Readability Score $\uparrow$ | 7.34($\pm$2.59)| **25.62($\pm$3.90)**  |
> || Readability Consensus Score $\uparrow$| 7.15($\pm$4.45)| **13.09($\pm$2.29)**  |
> || Spache Readability Formula $\uparrow$| 4.20($\pm$1.26)| **6.73($\pm$0.70)**|
>
> [1] [ChatGPT to Replace Crowdsourcing of Paraphrases for Intent Classification: Higher Diversity and Comparable Model Robustness](https://aclanthology.org/2023.emnlp-main.117) (Cegin et al., EMNLP 2023)
>
> **Questions**
>
> > **Regarding the question 1**
>
> Please refer to the response to Weakness 1.
>
>
> **Question 2 (Additional task)**
> > **Have you explored whether these patterns hold true for other tasks like toxicity detection or text classification? Even preliminary results would help assess generalizability.**
>
> We really appreciate your suggestions. Following your valuable guidance, we have supplemented the test of LLMs for **toxicity detection** using **[Jigsaw](https://www.kaggle.com/competitions/jigsaw-unintended-bias-in-toxicity-classification/overview)** as the dataset and present the results for the **maximum values of gender bias** below.
>
> In toxicity detection, LLMs are asked to judge whether the sentences given are toxic or non-toxic. In the table below, we provide the gender bias performance of *Llama-2-7B* and *Llama-3.2-3B* in toxicity detection. To stay clear, the results below are the maximum values of gender bias, and the remaining details (such as the mean values) are available in the revised appendix.
>
> |Llama-2-7B|$AvgGF$||$MaxTG$||$MaxFG$||
> |-|-|-|-|-|-|-|
> |$k=18$|Origin|ReBE|Origin|ReBE|Origin|ReBE|
> |Zero-shot|0.108|-|0.098|-|0.833|-|
> |Random-based|0.283|0.186|0.312|0.210|0.250|0.300|
> |Perplexity-based|0.205|0.168|0.217|0.173|0.667|0.667|
> |Similarity-based|0.154|0.141|0.140|0.129|0.500|0.667|
> |DPP-based|0.136|0.102|0.156|0.116|0.667|0.857|
>
> |Llama-3.2-3B|$AvgGF$||$MaxTG$||$MaxFG$||
> |-|-|-|-|-|-|-|
> |$k=18$|Origin|ReBE|Origin|ReBE|Origin|ReBE|
> |Zero-shot|0.145|-|0.158|-|0.429|-|
> |Random-based|0.215|0.108|0.217|0.127|0.500|0.550|
> |Perplexity-based|0.142|0.043|0.152|0.019|0.857|0.333|
> |Similarity-based|0.056|0.038|0.069|0.019|0.600|0.333|
> | DPP-based|0.090|0.048|0.049|0.011|0.750|0.500|
>
>
> Consistent with the sentiment analysis, **`we can find that:`**
>
> + Compared with the zero-shot, example selection methods for ICL **amplify the maximum value of gender bias**;
>
> - ReBE remains compatible with example selection methods and exhibits effective debiasing in toxicity detection.

---

> ### Author Response · Authors · 2024-11-24
> **Author Response - Part 2**
>
> **Weakness 2 (Baseline comparison)**
> > **The absence of comparisons with existing debiasing methods and fine-tuning approaches makes it difficult to fully assess ReBE's advantages.**
>
> Thank you for your valuable feedback. First, we would like to clarify that **fine-tuning methods are not suitable for comparison with ReBE**. Our work focuses on bias risks of example selections for ICL and debiasing methods that can serve ICL. However, fine-tuning approaches need to update the LLM parameters, which destroy the advantages of ICL.
>
> Second, **there are a few debiasing methods specifically for ICL** **`(Note that, they are NOT specifically for our research problem)`**. Although Hu et al. [1] proposed a fairness via clustering genetic (FCG) algorithm, FCG needs explicit feature vectors to complete the clustering. Due to this limitation, FCG cannot apply to sentiment analysis and toxicity detection, so we cannot set it as a baseline for ReBE. Even so, we value the reviewer's opinion and compare ReBE with two context augmentation methods: **Counterfactual** and **gender-balanced**.
>
> + **Counterfactual**
>
>   For a dataset built based on templates like *EEC*, it is convenient to construct the corresponding counterfactual instance according to the templates. For example, according to the template `<person subject> feels <emotion word>`, the counterfactual instance of sentence `Alonzo feels angry.` can be `Nichelle feels angry.` or `Amanda feels angry.`.
>
> + **Gender-balanced**
>
>   The gender-balanced context approach requires an equal or close number of examples for each gender type.
>
> To stay clear, we present the results of random-based and DPP-based example selections in the tables below. More details are available in the revised appendix.
>
> The following table shows the gender bias of *OPT-6.7B* on **Sentiment Analysis** with ***EEC-paraphrase*** as the dataset. While the *EEC-paraphrase* dataset does not have its own templates, it is built upon the *EEC* samples, allowing us to generate counterfactual samples using the templates from the *EEC*.
>
> |Sentiment Analysis|$AvgGF$(Mean)|Max|$MaxTG$(Mean)|Max|$MaxFG$(Mean)|Max|Acc|
> |-|-|-|-|-|-|-|-|
> |Random|0.044($\pm$0.03)|0.129|0.180($\pm$0.09)|0.468|0.199($\pm$0.09)|0.465|0.81|
> |DPP|0.036($\pm$0.03)|0.110|0.142($\pm$0.08)|0.273|0.144($\pm$0.06)|0.273|**0.87**|
> |Gender-balanced|0.040($\pm$0.03)|0.132|0.174($\pm$0.08)|0.333|0.210($\pm$0.09)|0.417|0.80|
> |Counterfactual|0.035($\pm$0.03)|0.125|0.145($\pm$0.07)|0.369|0.149($\pm$0.07)|0.369|0.77|
> |Random+ReBE|0.034($\pm$0.02)|0.086|0.151($\pm$0.07)|0.322|0.191($\pm$0.08)|0.447|0.78|
> |DPP+ReBE|**0.033($\pm$0.02)**|**0.073**|**0.120($\pm$0.05)**|**0.250**|**0.122($\pm$0.05)**|**0.247**|**0.87**|
>
> Since the counterfactual context method does not apply to datasets without templates,  we remove it from the baselines tested on the Jigsaw dataset. The following table shows the gender bias of *Llama-2-7B* on **Toxicity Detection** with ***Jigsaw*** as the dataset.
>
> |Toxicity Detection|$AvgGF$(Mean)|Max|$MaxTG$(Mean)|Max|$MaxFG$(Mean)|Max|Acc|
> |-|-|-|-|-|-|-|-|
> |Random|0.179($\pm$0.05)|0.283|0.215($\pm$0.05)|0.312|0.215($\pm$0.05)|0.312|0.76|
> |DPP|0.051($\pm$0.04)|0.136|0.059($\pm$0.04)|0.156|**0.171($\pm$0.18)**|0.667|0.85|
> |Gender-balanced|0.116($\pm$0.06)|0.236|0.205($\pm$0.08)|0.500|0.205($\pm$0.08)|0.500|0.81|
> |Random+ReBE|0.058($\pm$0.04)|0.186|0.070($\pm$0.04)|0.210|0.176($\pm$0.11)|**0.300**|0.86|
> |DPP+ReBE|**0.045($\pm$0.03)**|**0.102**|**0.053($\pm$0.02)**|**0.116**|0.248($\pm$0.20)|0.857|**0.88**|
>
> **`Takeaways:`**
>
> - There is currently **NO** suitable debiasing baseline specifically for ICL to compare with ReBE;
> - Compared with the counterfactual and gender-balanced context method, ReBE is compatible with existing example selection methods and can achieve **lower bias** and **higher accuracy**.
>
> [1] [Strategic Demonstration Selection for Improved Fairness in LLM In-Context Learning](https://aclanthology.org/2024.emnlp-main.425/) (Hu et al, EMNLP 2024)

---

> ### Author Response · Authors · 2024-11-27
> **A Kind Reminder**
>
> Dear Reviewer `Av69`,
>
> Thank you again for the valuable comments that helped improve our paper. Following your suggestions, we have carefully revised the manuscript to address your concerns.
>
> We kindly remind you to review our reply along with the revised submission and consider updating your evaluation accordingly.
>
> Best,
>
> Authors

---

> ### Author Response · Authors · 2024-12-03
> **A Kind Reminder**
>
> Dear Reviewer `Av69`,
>
> We appreciate all of the valuable time and effort you have spent reviewing our paper. As today is `the last day` of the discussion period, we gently request that you review our reply and consider updating your evaluation accordingly. We believe that we have addressed all questions and concerns raised, but please feel free to ask any clarifying questions you might have before the end of the discussion period.
>
> Best,
>
> Authors

---

### Official Review · Reviewer_wJ1K · 2024-11-03

**Soundness:** 2
**Presentation:** 1
**Contribution:** 2
**Rating:** 3
**Confidence:** 4

**Summary:**

This paper investigates how example selection methods for in-context learning (ICL) affect the biases of large language models (LLMs). The authors construct a new sentiment classification dataset, EEC-paraphrase, and discover three key findings: 1) high accuracy in example selection does not guarantee low bias, 2) example selection amplifies existing LLM biases, and 3) example selection contributes to spurious correlations in LLMs. To address these issues, they propose ReBE (Remind with Bias-aware Embedding), a method that uses contrastive learning and prompt tuning to mitigate biases while maintaining accuracy. The authors conduct extensive experiments using eight different LLMs and four example selection baselines. Their proposed ReBE method appears to reduce maximum bias values without compromising accuracy and demonstrates compatibility with existing example selection approaches.

**Strengths:**

- The paper raises an important and previously unexplored concern about how example selection might affect model bias.
- The experiments are well-structured across multiple models and methods, making the results more reliable.
- If the findings hold true, they have important implications for how ICL should be used in practice.

**Weaknesses:**

- The authors claim that their EEC-paraphrase dataset consists of sentences that are more complex and natural. However, there is no validation for this claim. The instructions to ChatGPT was to "expand the following sentence in more complex words." I'm skeptical that leads to sentences that are "more complex" and "natural". Furthermore, using an LLM to create sentences that are then used to study biases in other LLMs is not the right way to design this experiment.
- No ablation studies showing which components of ReBE are actually responsible for bias reduction. The paper claims ReBE "doesn't significantly compromise accuracy" but doesn't define what constitutes significant compromise. The authors haven't shared any statistical significance tests in the paper.
- The paper is not well written and the exposition needs a lot of improvement. Every figure and table should be clearly explained in text.
- All experiments are on a single task type (sentiment analysis). The authors haven't done any comparison to simpler bias mitigation approaches when many exist.

**Questions:**

- I'd like you to do some dataset validation, such as through human evaluation of the paraphrased sentences to ensure quality and meaning preservation, comparison with real-world text samples to validate ecological validity, and analysis of potential artifacts introduced by GPT-3.5 paraphrasing.
- I'd also like to see more methodological validation. Please include ablation studies isolating each component of ReBE. Share statistical significance testing for all reported results. I'd prefer to see some comparison with simpler debiasing approaches as well.
- Especially since you're using GPT-3.5 paraphrasing and not human annotation, could you come up with similar datasets for different types of tasks beyond sentiment analysis and conduct experiments there as well?
- There are many issues with the presentation. There is little explanation for figures or tables on how to read them. Then the writing itself has issues. Just as an example, why is Albaquerque et al the citation for Spurious correlations on Line 95? Then that is followed by, "Typical spurious correlations include stereotypes such as "He is a doctor; she is a nurse." That's not great academic writing. Please go over the paper with a red pen.

---

> ### Author Response · Authors · 2024-11-24
> **Author Response - Part 1**
>
> We sincerely appreciate your comments. Please find our point-by-point responses below:
>
> **Weakness 1 (Validation of *EEC-paraphrase*)**
> > **There is no validation for the claim that EEC-paraphrase dataset consists of sentences that are more complex and natural.**
>
> Thank you for your valuable feedback. To quantitatively demonstrate the complexity and naturalness of the *EEC-paraphrase* dataset, we evaluate using the Python library [Textstat](https://textstat.org/) and present the results in the table below.
>
> In selecting metrics, we choose the number of words and Distinct-n to measure the sentence diversity, which reflects the **naturalness**. To evaluate **complexity**, we use metrics from Textstat to determine the readability, complexity, and grade level of sentences.
>
> The results show that the *EEC-paraphrase* dataset **`outperforms`** the original *EEC* dataset across **all** metrics.
>
> ||**Metric**|**EEC**|**EEC-paraphrase**|
> |-|--|-|-|
> |**Diversity**|Numberofwords$\uparrow$|5.86($\pm$1.73)|**18.63($\pm$2.33)**|
> ||Distinct-2$\uparrow$|0.81($\pm$0.066)|**0.94($\pm$0.147)**|
> ||Distinct-3 $\uparrow$| 0.62($\pm$0.132)| **0.89($\pm$0.017)**|
> |**Complexity** |Automated Readability Index $\uparrow$|7.56($\pm$3.80)|**14.44($\pm$2.27)**|
> ||Coleman-Liau Index $\uparrow$| 9.63($\pm$4.57)|**14.74($\pm$2.92)**|
> ||Dale-Chall Readability Score $\uparrow$| 11.94($\pm$3.21)| **11.68($\pm$1.17)**|
> ||Flesch-Kincaid Grade Level $\uparrow$| 5.52($\pm$3.68)| **12.06($\pm$2.04)**|
> ||Flesch Reading Ease Score $\downarrow$| 65.88($\pm$26.09)|**41.68($\pm$14.38)**|
> ||Fog Scale $\uparrow$| 8.44($\pm$5.04)| **15.18($\pm$2.88)**|
> ||Linsear Write Formula $\uparrow$| 2.87($\pm$1.25)| **12.83($\pm$2.06)**|
> ||McAlpine EFLAW Readability Score $\uparrow$ | 7.34($\pm$2.59)| **25.62($\pm$3.90)**|
> ||Readability Consensus Score $\uparrow$| 7.15($\pm$4.45)| **13.09($\pm$2.29)**|
> ||Spache Readability Formula $\uparrow$| 4.20($\pm$1.26)| **6.73($\pm$0.70)**|
>
> > **Using an LLM to create sentences that are then used to study biases in other LLMs is not the right way to design this experiment.**
>
> Thanks for your valuable feedback. First, we argue that the statement "Using an LLM to create sentences that are then used to study biases in other LLMs" **is not accurate** because *EEC-paraphrase* is not created out of thin air by *GPT-3.5-Turbo*, but paraphrase the existing dataset *EEC*.
>
> Second, **previous work has demonstrated the capability of LLMs (including GPT-3.5-Turbo) to produce diverse and valid paraphrases under guidance**[1]. Therefore, we think that leveraging LLMs, **under human review**, to assist in part of the dataset construction process—rather than relying on crowdsourcing—is a reasonable experimental design. **We follow the practices adopted by previous work**.
>
> [1] ChatGPT to Replace Crowdsourcing of Paraphrases for Intent Classification: Higher Diversity and Comparable Model Robustness (Cegin et al., EMNLP 2023)
>
> **Weakness 3**
> > **The paper is not well written and the exposition needs a lot of improvement.**
>
> Thank you for your valuable feedback. We have tried our best to polish the language in the revised manuscript. We would be grateful if you could provide more specific examples.
>
> **Weakness 4.1 (Additional task)**
> > **All experiments are on a single task type (sentiment analysis).**
>
> Thank you for your valuable feedback. We have supplemented the test of LLMs for **toxicity detection** using **[Jigsaw](https://www.kaggle.com/competitions/jigsaw-unintended-bias-in-toxicity-classification/overview)** as the dataset and present the results for the **maximum values of gender bias** below.
>
> In toxicity detection, LLMs are asked to judge whether the sentences given are toxic or non-toxic. In the table below, we provide the gender bias performance of *Llama-2-7B* and *Llama-3.2-3B* in toxicity detection. To stay clear, the results below are the maximum values of gender bias, and the remaining details (such as the mean values) are available in the revised appendix.
>
> |Llama-2-7B|$AvgGF$||$MaxTG$||$MaxFG$||
> |-|-|-|-|-|-|-|
> |$k=18$|Origin|ReBE|Origin|ReBE|Origin|ReBE|
> |Zero-shot|0.108|-|0.098|-|0.833|-|
> |Random-based|0.283|0.186|0.312|0.210|0.250|0.300|
> |Perplexity-based|0.205|0.168|0.217|0.173|0.667|0.667|
> |Similarity-based|0.154|0.141|0.140|0.129|0.500|0.667|
> |DPP-based|0.136|0.102|0.156|0.116|0.667|0.857|
>
> |Llama-3.2-3B|$AvgGF$||$MaxTG$||$MaxFG$||
> |-|-|-|-|-|-|-|
> |$k=18$|Origin|ReBE|Origin|ReBE|Origin|ReBE|
> |Zero-shot|0.145|-|0.158|-|0.429|-|
> |Random-based|0.215|0.108|0.217|0.127|0.500|0.550|
> |Perplexity-based|0.142|0.043|0.152|0.019|0.857|0.333|
> |Similarity-based|0.056|0.038|0.069|0.019|0.600|0.333|
> | DPP-based|0.090|0.048|0.049|0.011|0.750|0.500|
>
> Consistent with the sentiment analysis, **we can find that:**
>
> - Compared with the zero-shot, example selection methods for ICL **amplify the maximum value of gender bias**;
> - ReBE remains compatible with example selection methods and exhibits effective debiasing in toxicity detection.

---

> ### Author Response · Authors · 2024-11-24
> **Author Response - Part 2**
>
> > **[W4.2] The authors haven't done any comparison to simpler bias mitigation approaches when many exist.**
>
> Thank you for your valuable feedback. We would like to clarify that we have presented a discussion of other debiasing methods in Subsection 3.5 (Appendix G.1 of the current revision).
>
> First, **there are a few debiasing methods specifically for ICL (Note that, they are NOT specifically for our research problem)**. Although Hu et al. [1] proposed a fairness via clustering genetic (FCG) algorithm, FCG needs explicit feature vectors to complete the clustering. Due to this limitation, FCG cannot apply to sentiment analysis and toxicity detection, so we cannot set it as a baseline for ReBE. Second, we value the reviewer's opinion and compare ReBE with two context augmentation methods: **Counterfactual** and **gender-balanced**.
>
> + **Counterfactual**
>
>   For a dataset built based on templates like *EEC*, it is convenient to construct the corresponding counterfactual instance according to the templates. For example, according to the template `<person subject> feels <emotion word>`, the counterfactual instance of sentence `Alonzo feels angry.` can be `Nichelle feels angry.` or `Amanda feels angry.`.
>
> + **Gender-balanced**
>
>   The gender-balanced context approach requires an equal or close number of examples for each gender type.
>
> To stay clear, only the results of random-based example selection are shown below. More details are available in the revised appendix.
>
> The following table shows the gender bias of *OPT-6.7B* on **Sentiment Analysis** with ***EEC-paraphrase*** as the dataset. While the *EEC-paraphrase* dataset does not have its own templates, it is built upon the *EEC* samples, allowing us to generate counterfactual samples using the templates from the *EEC*.
>
> |Sentiment Analysis|$AvgGF$(Mean)|Max|$MaxTG$(Mean)|Max|$MaxFG$(Mean)|Max|Acc|
> |-|-|-|-|-|-|-|-|
> |Random|0.044($\pm$0.03)|0.129|0.180($\pm$0.09)|0.468|0.199($\pm$0.09)|0.465|0.81|
> |DPP|0.036($\pm$0.03)|0.110|0.142($\pm$0.08)|0.273|0.144($\pm$0.06)|0.273|**0.87**|
> |Gender-balanced|0.040($\pm$0.03)|0.132|0.174($\pm$0.08)|0.333|0.210($\pm$0.09)|0.417|0.80|
> |Counterfactual|0.035($\pm$0.03)|0.125|0.145($\pm$0.07)|0.369|0.149($\pm$0.07)|0.369|0.77|
> |Random+ReBE|0.034($\pm$0.02)|0.086|0.151($\pm$0.07)|0.322|0.191($\pm$0.08)|0.447|0.78|
> |DPP+ReBE|**0.033($\pm$0.02)**|**0.073**|**0.120($\pm$0.05)**|**0.250**|**0.122($\pm$0.05)**|**0.247**|**0.87**|
>
> Since the counterfactual context method does not apply to datasets without templates,  we remove it from the baselines tested on the Jigsaw dataset. The following table shows the gender bias of *Llama-2-7B* on **Toxicity Detection** with ***Jigsaw*** as the dataset.
>
> |Toxicity Detection|$AvgGF$(Mean)|Max|$MaxTG$(Mean)|Max|$MaxFG$(Mean)|Max|Acc|
> |-|-|-|-|-|-|-|-|
> |Random|0.179($\pm$0.05)|0.283|0.215($\pm$0.05)|0.312|0.215($\pm$0.05)|0.312|0.76|
> |DPP|0.051($\pm$0.04)|0.136|0.059($\pm$0.04)|0.156|**0.171($\pm$0.18)**|0.667|0.85|
> |Gender-balanced|0.116($\pm$0.06)|0.236|0.205($\pm$0.08)|0.500|0.205($\pm$0.08)|0.500|0.81|
> |Random+ReBE|0.058($\pm$0.04)|0.186|0.070($\pm$0.04)|0.210|0.176($\pm$0.11)|**0.300**|0.86|
> |DPP+ReBE|**0.045($\pm$0.03)**|**0.102**|**0.053($\pm$0.02)**|**0.116**|0.248($\pm$0.20)|0.857|**0.88**|
>
> **`Takeaways:`**
>
> - There is currently **NO** suitable debiasing baseline specifically for ICL to compare with ReBE;
> - Compared with the counterfactual and gender-balanced context method, ReBE is compatible with existing example selection methods and can achieve **lower bias** and **higher accuracy**.
>
> [1] [Strategic Demonstration Selection for Improved Fairness in LLM In-Context Learning](https://aclanthology.org/2024.emnlp-main.425/) (Hu et al, EMNLP 2024)
>
> **Questions:**
>
> > **[Q1] I'd like you to do some dataset validation, such as through human evaluation of the paraphrased sentences to ensure quality and meaning preservation, comparison with real-world text samples to validate ecological validity, and analysis of potential artifacts introduced by GPT-3.5 paraphrasing.**
>
> Please refer to the response to Weakness 1.
>
>
>
> > **[Q2] Please include ablation studies isolating each component of ReBE. I'd prefer to see some comparison with simpler debiasing approaches as well.**
>
> Please refer to the response to Weakness 2.1 and Weakness 4.2.
>
>
>
> > **[Q3] Could you come up with similar datasets for different types of tasks beyond sentiment analysis and conduct experiments there as well?**
>
> Please refer to the response to Weakness 4.1.
>
>
>
> > **[Q4] There are many issues with the presentation.**
>
> Please refer to the response to Weakness 3.

---

> ### Author Response · Authors · 2024-11-24
> **Author Response - Part 3**
>
> **Weakness 2 (Ablation study and significance test)**
> > **No ablation studies showing which components of ReBE are actually responsible for bias reduction.**
>
> Thank you for your valuable feedback. We would like to clarify that we included the **ablation study in subsection 5.2** and provided the **experiment results in Table 4**.
>
> As for the components responsible for bias reduction, since ReBE introduces contrastive loss based on prompt tuning, components that can be used for ablation experiments are the loss functions.
>
> In subsection 5.2, we tested the cases of $L_{acc}$ and $L_{bias}$, which represent the cases of not using $L_{bias}$ and not using $L_{acc}$, respectively. Based on the results in Table 4, we have revised and concluded that **$L_{bias}$ is actually responsible for bias reduction**, and $L_{acc}$ guarantees accuracy (lines 444-446).
>
>
>
> > **The authors haven't shared any statistical significance tests in the paper.**
>
> Thank you for your valuable feedback. We would like to clarify that statistical significance test is not suitable for verification of results in this paper.
>
> The statistical significance test can help determine whether there is a difference between two data sets and whether the difference is significant. In other words, it estimates **the probability that two data sets belong to the same overall distribution**. **However**, because the inputs of LLM in ICL are constructed based on the same dataset and belong to the same distribution, the outputs of the same LLM under various example selection methods still belong to the same overall distribution. Therefore, **the statistical significance test is not applicable to the comparison between different example selection methods.**
>
> *Why do the inputs belong to the same distribution ?*
>
> In ICL, each input consists of a question and a context, which is a combination of $k$ samples. Although different example selection methods select different $k$ samples to construct the context, these samples belong to the same dataset. When the context distribution of random-based example selection include enough points, other example selection methods can be understood as sampling part of this distribution according to specific rules. So, theoretically, inputs of LLM under various example selection methods belong to the same overall distribution.
>
> *Why is the statistical significance test also not applicable to the comparison of **ReBE** ?*
>
> Since ReBE is implemented based on prompt tuning without updating LLM parameters, the above discussion on output distribution still applies to the combination of ReBE and other example selection methods. More specifically, ReBE adds a small amount of virtual tokens to the original input. Compared with the input length (1024) of LLMs, the number of virtual tokens is tiny (10-30), so the output distribution of ReBE could be close to the example selection methods, **the statistical significance test is not applicable to the comparison between ReBE and other example selection methods.**
>
> *What did we do to ensure the reliability of the results ?*
>
> To reduce the randomness of results, we followed the advice and tested nine LLMs on two tasks (sentiment analysis and toxicity detection) under multiple random seeds.

---

> ### Author Response · Authors · 2024-11-27
> **A Kind Reminder**
>
> Dear Reviewer `wJ1K`,
>
> Thank you again for the valuable comments that helped improve our paper. Following your suggestions, we have carefully revised the manuscript to address your concerns.
>
> We kindly remind you to review our reply along with the revised submission and consider updating your evaluation accordingly.
>
> Best,
>
> Authors

---

> > ### Comment · Reviewer_wJ1K · 2024-11-28
> > **Thanks for the response. Some concerns addressed but most remain.**
> >
> > Thank you for your detailed response to my review. While your additional analyses and explanations help address some concerns, I believe several important issues still need to be addressed to strengthen the paper.
> >
> > Regarding the EEC-paraphrase dataset, the quantitative metrics you've provided demonstrate increased complexity but don't fully validate the dataset's quality. Consider your example paraphrase: "Alan is experiencing a profound sense of frustration and irritation, resulting in a heightened state of emotional turmoil and discomfort." While this scores higher on complexity metrics, it contains redundant phrasing and potentially unnatural language. ("frustration and irritation," "turmoil and discomfort").
> >
> > This illustrates why we need human evaluation to verify: i) Meaning preservation - Does the paraphrase maintain the original sentiment and intensity?, ii) Natural language use - Would humans actually express the emotion this way?, iii) Potential artifacts - Are there systematic patterns in how GPT-3.5 paraphrases that could bias the results? I strongly recommend conducting human evaluation to verify meaning preservation, naturalness, and potential systematic biases introduced by GPT-3.5's paraphrasing patterns.
> >
> > The toxicity detection experiments are a valuable addition, but they raise important questions about the consistency of ReBE's effectiveness across tasks. The substantial difference in MaxFG values between sentiment analysis (0.2-0.4) and toxicity detection (up to 0.857) suggests task-dependent performance that warrants deeper analysis and discussion in the paper.
> >
> > Regarding ablation studies, while Table 4's analysis of loss functions is helpful no doubt, more granular investigation would strengthen the work. This could include analyzing different contrastive learning configurations (positive/negative pair construction, temperature parameter tuning, similarity metrics), prompt tuning variations (length, position, architecture), and comprehensive loss function analysis (weight parameter sweeps, alternative formulations, training dynamics). More on this in a separate comment.
> >
> > Your explanation regarding statistical significance testing, while thoughtful, doesn't address the need to validate the reliability of your results. While the inputs may come from related distributions, we need to verify that performance differences between methods are meaningful rather than random variation. Standard approaches like paired t-tests or bootstrap resampling would help establish the robustness of your findings. "Because the inputs of LLM in ICL are constructed based on the same dataset and belong to the same distribution, the outputs of the same LLM under various example selection methods still belong to the same overall distribution." This reasoning is flawed because we're not interested in whether the outputs come from the same distribution---we want to know if the differences in performance metrics (accuracy, bias measures) between methods are statistically reliable.
> >
> > Finally, the paper's presentation would benefit from clearer figure explanations (particularly Figure 1's crucial "grey area"), better motivation of methodological choices, and a more cohesive literature review that builds a narrative rather than listing related work. The methodology section should better justify choices like: i) The specific contrastive learning formulation, ii) The choice of loss function components, iii) The prompt tuning architecture. The literature review currently presents related work as a list of papers rather than building a coherent narrative about how the field has developed and where this work fits.

---

> > > ### Comment · Reviewer_wJ1K · 2024-11-28
> > > **Regarding Ablation Studies**
> > >
> > > The current Table 4 has several limitations that prevent it from definitively attributing performance gains to ReBE.
> > >
> > > First, comparing just $L_acc$, $L_bias$, and ReBE's combined loss does not isolate all of ReBE's components. ReBE introduces both a contrastive learning mechanism and a prompt tuning approach. The current ablation simply shows the impact of different loss functions while keeping the rest of the architecture constant. To properly validate ReBE's effectiveness, we would need to test each architectural component independently.
> > >
> > > Second, the table presents aggregate metrics without showing how different components interact. For example, we cannot tell if the contrastive learning is truly removing spurious correlations or if the improved metrics come from other aspects of the model. A more comprehensive ablation would track specific types of biases and show how each component affects them.
> > >
> > > For instance, you identify a specific type of spurious correlation in Figure 3, showing that sentences labeled as "sadness" containing male pronouns are more frequently misclassified as "fear" compared to those with female pronouns (0.54 vs 0.08). You then claim that ReBE helps remove such spurious correlations through its contrastive learning component. However, Table 4's ablation study only shows overall metrics (Accuracy, AvgGF, MaxTG, MaxFG) when using different loss functions. What's missing is a direct connection between these metrics and the specific spurious correlations the paper identified. We need to see: i) How the male-sadness-to-fear misclassification rate specifically changes when using just $L_acc$ versus just $L_bias$ versus the full ReBE model, ii) Whether the contrastive learning component is actually learning to separate these specific cases, or if the improved metrics come from other effects of the architecture, iii) A demonstration that the positive/negative pairs in the contrastive learning are effectively capturing and correcting these specific biased associations.
> > >
> > > Without this level of analysis, we cannot verify whether ReBE is truly addressing the spurious correlations it claims to target, or if the improved bias metrics are coming from other aspects of the model architecture. This distinction matters because it affects both our understanding of the problem and our confidence in the proposed solution.
> > >
> > > A more rigorous ablation study would systematically remove or modify each component of ReBE while measuring its impact on both performance and different types of bias (as described above). This would provide stronger evidence for which parts of the architecture are responsible for the observed improvements.

---

> ### Author Response · Authors · 2024-11-29
> **Clarification of some misunderstandings**
>
> Thank you for your valuable feedback. First, we think **`we have addressed and responded to most of the concerns in our rebuttal above`**. Summarize them as follows.
>
> > **EEC-paraphrase dataset validation**
>
> We have addressed it. Please refer to $\rightarrow$ Author Response - Part 1: Weakness 1 (Validation of EEC-paraphrase)
>
> > **Ablation studies**
>
> We have addressed it. Please refer to $\rightarrow$ Author Response - Part 3: Weakness 2 (Ablation study and significance test)
>
> Second, we would like to provide further explanations for some **misunderstandings**.
>
> > **The substantial difference in MaxFG values between sentiment analysis (0.2-0.4) and toxicity detection (up to 0.857) suggests task-dependent performance.**
>
> It is `unreasonable` to compare the debiasing results without considering the `original data` and `task differences`.
>
> > **This reasoning is flawed because we're not interested in whether the outputs come from the same distribution---we want to know if the differences in performance metrics (accuracy, bias measures) between methods are statistically reliable.**
>
> Because the **performance metrics are calculated completely based on the outputs**, we think the reasoning is valid.
>
> > **Some suggestions for improving presentations**
>
> Thank you for your valuable suggestions; since the reviewer's latest reply is after the revision submission deadline, we are unable to upload the improved manuscript according to the suggestions. However, we will include the corresponding revisions in subsequent updates.
>
> Thank you again for your valuable feedback!

---

> > ### Comment · Reviewer_wJ1K · 2024-12-02
> > **Thanks again for your response**
> >
> > While I appreciate the authors' response and attempts and addressing my concerns, I don't believe they have been addressed and my score will reflect the same.

---

> ### Author Response · Authors · 2024-12-02
> **Thanks for your reply, and more specific points please?**
>
> Dear Reviewer wJ1K,
>          Thank you for your reply.  We do believe our rebuttal has addressed the concerns you posted in your comments. We also clarified some misunderstandings that you may have.  For a better discussion, **could you please clearly specify the specific technical points of your concerns that you have?**   Greatly appreciate that!

---

### Official Review · Reviewer_CqEA · 2024-11-04

**Soundness:** 3
**Presentation:** 3
**Contribution:** 3
**Rating:** 6
**Confidence:** 3

**Summary:**

The paper introduces Remind with Bias-aware Embedding (ReBE), a method aimed at mitigating bias in LLMs by addressing spurious correlations through prompt tuning. Using a newly constructed version of Equity Evaluation Corpus, the authors evaluate how in-context learning (ICL) example selection influence biases related to gender and race. This paper considers a variety of GPT, OPT, and Llama models and study bias amplification using several prompt selection techniques: Random example selection, Perplexity, Similarity and DPP. Results show that ICL prompt selection generally does not increase average bias regardless of the model/example selection method, as measured by Average Group Fairness, it can amplify maximum bias levels measured by Maximum TPR Gap and Maximum FPR Gap.  In order to reduce the increase in maximum bias levels caused by ICL example selection, the authors introduce ReBE. ReBE is designed using a contrastive loss function that encourages bias-aware embeddings, aiming to reduce biases without significantly impacting model accuracy.

**Strengths:**

1. Novelty in Addressing Bias in ICL Example Selection: The paper tackles an underexplored problem, focusing on how example selection of ICL prompts with example selection amplify bias in LLMs. Demonstrating that different example selection methods increase maximum bias is an important finding. Further, disentangling native bias in the parameters of the model with ICL example selection bias provides a more holistic evaluation of bias in LLM outputs.

2. Effectiveness of ReBE: Figure 7 demonstrates that ReBE reduces bias in LLMs more than several other example selection methods as measured by Max FG. Further, the accuracy of using DPP + ReBE does not appear to impact the accuracy of the LLM significantly.

3. Comprehensive Experimental Setup: The authors conduct experiments across multiple model types and sizes (e.g., LLaMA-2, OPT), which strengthens the generalizability of the findings.

**Weaknesses:**

1. The relationship between ICL example selection and bias amplification is complex and not as straightforward as the authors present it. Figure 2 demonstrates that example selection reduces average bias across most models. The claim that “all LLMs exhibit an increase in the maximum gender or race bias value with random-based example selection for ICL” (lines 217-218). This statement may be too bold of a claim based on their mixed results of bias amplification.

2. $\textbf{ReBE does not always significantly reduce bias}$: While ReBE reduces bias in many cases, its improvements over DPP are limited. Figure 7 shows that the decrease in bias is modest, which may raise questions about ReBE’s overall efficacy.

3. There is no Analysis on the impact of increasing the number of ICL examples. Conducting bias analysis for ICL example selection at a different number of shots for all ICL example selection methods in this paper would be an important addition to the paper. Some analysis of the impact of the number of shots is conducted in Figure 8, but this is limited only to ReBE.

**Questions:**

1. Although ReBE is presented in the paper, Figure 6 is a little confusing and is not fully explained either in the main body of the paper or the Figure caption.

2. How does increasing the number of ICL examples impact bias in other example selection methods?

3. Would the authors be willing to elaborate on the additional train time required to add ReBE to existing methods?

---

> ### Author Response · Authors · 2024-11-24
> **Author Response - Part 1**
>
> We thank the reviewer for your recognition of our work. We also appreciate the detailed comments posed by the reviewer. Please find below the point-to-point responses to your comments.
>
> **Weakness 1 (The claim may be too bold)**
> > **The claim that “all LLMs exhibit an increase in the maximum gender or race bias value with random-based example selection for ICL” (lines 217-218) may be too bold of a claim based on their mixed results of bias amplification.**
>
> Thank you for your valuable feedback. To make the wording more rigorous, we have corrected the claim as "the LLMs tested exhibit varying degrees of increase in the maximum gender or race bias value with random-based example selection for ICL" (lines 223-224).
>
> Besides, we would like to give a brief explanation as to whether the claim is too bold. Although some LLMs do not exhibit **simultaneous** amplification of the maximum values of gender and race bias in sentiment analysis, we do not argue that the claim does not apply to these models. This is because the experiment results are task-dependent, and our additional experiment results for toxicity detection support this.
>
> For instance, in sentiment analysis, *Llama-2-7B* does not show an amplification of the maximum value of gender bias, but it does exhibit significant gender bias in toxicity detection (Detailed data is available in the revision appendix). Therefore, as long as the LLM shows amplification of the maximum value of a single type of bias, we think the claim applies to it.
>
> **Weakness 2 (Limited bias mitigation over DPP)**
> > **While ReBE reduces bias in many cases, its improvements over DPP are limited (Figure 7).**
>
> Thank you for your valuable feedback. Firstly, we acknowledge that reducing bias becomes increasingly challenging when the initial level of bias is already minimal. Compared with other example selection methods, the bias values of DPP-based example selection are smaller (The **max value** of *AvgGF* is just 7% in Figure 6, which is the Figure 7 of the previous submission). Additionally, it is generally understood that the goal of a debiasing method is to reduce bias to an acceptable range, as completely eliminating it is often difficult to achieve.
>
> **Weakness 3 (Impact of $k$)**
> > **There is no analysis on the impact of increasing the number of ICL examples.**
>
> Thank you for your suggestions. We have added the corresponding experiment results and analysis in **Appendix E**. To investigate the impact of increasing the number of IC examples, we have assessed the gender bias performance of  ***Llama-2-7B*** in toxicity detection under various number of ICL examples ($k\in[2,6,10,14,18,22,26]$).
>
> 1. Gender bias performance of *Llama-2-7B* on $AvgGF$
>
> || Random|| Perplexity|| Similarity|| DPP||
> | :- | :-: | :---: | :-: | :---: | :-: | :---: | :-: | :---: |
> || Mean| Max| Mean| Max| Mean| Max| Mean| Max|
> | $k=2$  | 0.204($\pm$0.05) | 0.310 | 0.105($\pm$0.05) | 0.195 | 0.116($\pm$0.05) | 0.203 | 0.129($\pm$0.05) | 0.235 |
> | $k=6$  | 0.204($\pm$0.05) | 0.312 | 0.087($\pm$0.06) | 0.211 | 0.067($\pm$0.04) | 0.166 | 0.073($\pm$0.05) | 0.195 |
> | $k=10$ | 0.175($\pm$0.03) | 0.247 | 0.075($\pm$0.06) | 0.187 | 0.046($\pm$0.03) | 0.158 | 0.062($\pm$0.04) | 0.156 |
> | $k=14$ | 0.179($\pm$0.05) | 0.263 | 0.083($\pm$0.05) | 0.189 | 0.041($\pm$0.03) | 0.127 | 0.056($\pm$0.03) | 0.105 |
> | $k=18$ | 0.179($\pm$0.05) | 0.283 | 0.058($\pm$0.06) | 0.205 | 0.043($\pm$0.05) | 0.154 | 0.051($\pm$0.04) | 0.136 |
> | $k=22$ | 0.187($\pm$0.05) | 0.285 | 0.064($\pm$0.06) | 0.211 | 0.035($\pm$0.03) | 0.109 | 0.041($\pm$0.03) | 0.094 |
> | $k=26$ | 0.151($\pm$0.04) | 0.229 | 0.063($\pm$0.05) | 0.198 | 0.038($\pm$0.03) | 0.130 | 0.046($\pm$0.02) | 0.078 |
>
> 2. Gender bias performances of *Llama-2-7B* on $MaxTG$ and $MaxFG$ are available in the revision appendix.
>
> **Findings:** As the number of ICL examples $k$ increases, the bias decreases overall, but the change in accuracy must also be considered. Please see the revision for detailed additional data.
>
> **Question 1 (Confusing Figure 6)**
> > **Figure 6 is a little confusing and is not fully explained either in the main body of the paper or the Figure caption.**
>
> We apologize for the confusion. We have added more detailed explanation and hope this can resolve your confusion (lines 342-345).
>
> **Question 2 (Impact of $k$)**
> > **How does increasing the number of ICL examples impact bias in other example selection methods?**
>
> Please refer to the **findings** in the response to Weakness 3.
>
>
> **Question 3 (Additional train time)**
>
> > **The additional train time required to add ReBE to existing methods.**
>
> That's a good question, we give an example in the following table.
>
> | Task| Model| Dataset | Train set size | Dev set size | epochs | GPU| Time consuming |
> | :-: | :-: | :-: | :-: | :-: | :-: | - | :-: |
> | Toxicity Detection | Llama-2-7B | Jigsaw  | 400| 200| 20| A100 80GB | `2.5h`|

---

> > ### Comment · Reviewer_CqEA · 2024-11-25
> > **Response**
> >
> > Thank you for your thoughtful and detailed response. The results presented regarding the number of ICL examples help strengthen your paper. After reading the other reviews and comments, I will keep my score the same.

---

### Author Response · Authors · 2024-11-24
**General Response: Novelties and Answers to common concerns**

Dear Chairs and Reviewers,

We sincerely thank all the reviewers for their time and valuable feedback on our work.

In this work, **we are the first** to explore **the `severe bias risks` of example selection methods for ICL** , which is **`ignored by previous work`**, and **propose a novel debiasing method, ReBE.** Unlike fine-tuning, ICL is more flexible and suitable for few-shot scenarios, as it requires minimal data and avoids parameter updates. However, utilizing ICL to deploy LLMs to downstream tasks **has the risk of preserving or even exacerbating biases.** Therefore, we try to **`fill this gap`** by exploring and mitigating the ethical risks of example selection, **which is `really non-trivial`**.

We summarize the **common concerns** raised by reviewers and **our responses** as follows:

1. **Lack of the *EEC-paraphrase* dataset validation**

   **Response**: Previous work has demonstrated the capability of LLMs (including GPT-3.5-Turbo) to produce diverse and valid paraphrases under guidance [1]. To verify the quality and diversity of *EEC-paraphrase*, we conduct the evaluation of *EEC-paraphrase* on various metrics and find that *EEC-paraphrase* **outperforms** the original *EEC* dataset across all metrics. We also manually sampled and reviewed the sentences in *EEC-paraphrase*.    **Our dataset validation process follows the practices adopted by previous work.**

2. **Lack of tasks beyond the sentiment analysis**

   **Response**: To verify the generalizability, we have **supplemented the test** of LLMs for **toxicity detection** using Jigsaw as the dataset. Consistent with the results in sentiment analysis, example selection methods for ICL **amplify the maximum value of gender bias**, and **ReBE also exhibits effective debiasing in toxicity detection**.

3. **Lack of comparison between ReBE and debiasing baselines**

   **Response**: **Our paper is the first work** to discover the bias problem, and thus no baseline could solve the problem we found. Since there is currently no suitable debiasing baseline specifically for ICL, we compare ReBE with two context augmentation methods. Compared with the context counterfactual and gender-balanced method, ReBE is compatible with existing example selection methods and can achieve **lower bias** and **higher accuracy**.

[1] [ChatGPT to Replace Crowdsourcing of Paraphrases for Intent Classification: Higher Diversity and Comparable Model Robustness](https://aclanthology.org/2023.emnlp-main.117) (Cegin et al., EMNLP 2023)


Other valuable comments from the reviewers are responded to point-by-point below.


Below, we summarize **`a list of revisions made in the newest updated submission`** for your review.

> **Section 1**
>
> + **Lines 94-95**: Corrected the location of the inserted reference.

> **Section 2**
>
> + **Lines 174-175**: Added the citation to dataset validation of Appendix A.
> + **Lines 194-195**: Added the citation to toxicity detection experiments of Appendix F.
> + **Lines 223-225**: Stated the finding more rigorously.

> **Section 4**
>
> + **Lines 342-344**: Added more detailed explanation of Figure 5.

> **Section 5**
>
> + **Lines 444-446**: Explicitly illustrated the results of the ablation study.
> + **Lines 460-476**: Added the baseline comparison of ReBE.
> + **Lines 486-487**: Added the citation to supplementary experiments on the impact of  $k$ on bias.
>
> **Appendix**
>
> + **Appendix A.1 (Lines: 717-740)**: Added the description and results of dataset validation.
> + **Appendix E**: Added a new section to analyze the impact of $k$ on bias.
> + **Appendix F**: Added the experiments on toxicity detection.
> + **Appendix G**: Added the baseline comparison of ReBE.
> + **Appendix G.1**: Moved the debiasing discussion previously in subsection 3.5 to the newly added Appendix G.1 for better organization.

Thank you again for your constructive comments.

Kind regards,

The authors

---

### Meta-Review · Area_Chair_9oQm · 2024-12-22

**Metareview:**

After carefully reviewing both the manuscript and the authors' rebuttal, I find that while this paper addresses an important and novel problem regarding bias amplification in example selection for in-context learning, and the overall research direction shows promise, as the reviewers have pointed out, the current manuscript has several methodological and experimental limitations that need to be addressed. The reviewers have raised valid concerns about the dataset construction methodology, the completeness of the experimental evaluation, and some inconsistencies in the reported results. The paper would benefit from more rigorous empirical validation, broader comparisons with existing approaches, and clearer demonstration of the proposed method's effectiveness, some of which the authors have begun to address in their rebuttal. Given these substantial concerns, I recommend that the authors carefully incorporate the reviewers' detailed feedback to strengthen their methodology and experiments, and consider submitting a revised version to a future venue. With these improvements, particularly in the experimental rigor and empirical validation, this work has the potential to make valuable contributions to our understanding of bias in large language models.

**Additional Comments On Reviewer Discussion:**

I have read the messages in the discussion period and my opinion has been summarized as in the metareview above. I considered these points in my recommendation.

---

### Decision · Program_Chairs · 2025-01-22

Reject